# Transneuronal delivery of hyper-interleukin-6 enables functional recovery after severe spinal cord injury in mice

Marco Leibinger[1], Charlotte Zeitler[1], Philipp Gobrecht[1], Anastasia Andreadaki[1], Günter Gisselmann[1] & Dietmar Fischer [1 ✉]

Spinal cord injury (SCI) often causes severe and permanent disabilities due to the regenerative failure of severed axons. Here we report significant locomotor recovery of both hindlimbs after a complete spinal cord crush. This is achieved by the unilateral transduction of cortical motoneurons with an AAV expressing hyper-IL-6 (hIL-6), a potent designer cytokine stimulating JAK/STAT3 signaling and axon regeneration. We find collaterals of these AAV-transduced motoneurons projecting to serotonergic neurons in both sides of the raphe nuclei. Hence, the transduction of cortical neurons facilitates the axonal transport and release of hIL-6 at innervated neurons in the brain stem. Therefore, this transneuronal delivery of hIL-6 promotes the regeneration of corticospinal and raphespinal fibers after injury, with the latter being essential for hIL-6-induced functional recovery. Thus, transneuronal delivery enables regenerative stimulation of neurons in the deep brain stem that are otherwise challenging to access, yet highly relevant for functional recovery after SCI.

---

[1] Department of Cell Physiology, Ruhr University of Bochum, Universitätsstraße 150, 44780 Bochum, Germany. ✉email: dietmar.fischer@rub.de

Neurons of the adult mammalian central nervous system (CNS) do not naturally regenerate injured axons. This regenerative failure often causes severe and permanent disabilities, such as para- or tetraplegia after spinal cord injury. To date, no cures are available in the clinic, underscoring the need for novel therapeutic strategies enabling functional recovery in respective patients.

Besides an inhibitory environment for axonal growth cones caused by myelin or the forming glial scar, the lack of CNS regeneration is mainly attributed to a developmental decline in the neuron-intrinsic growth capacity of axons per se[1–3]. Among all descending pathways, the corticospinal tract (CST), which controls voluntary fine movements, is the most resistant to regeneration. Despite numerous efforts aiming to facilitate the CST's axon regrowth over the last decades, such as the delivery of neurotrophic factors[4–6] or neutralizing inhibitory cues[7–9], success has remained very limited. However, the conditional genetic knockout of the phosphatase and tensin homolog (PTEN$^{-/-}$) in cortical motor neurons, which leads to an activation of the phosphatidylinositol-3-kinase/protein kinase B (PI3K/AKT)/mTOR signaling pathway, enables some regeneration of CST axons beyond the site of injury[10]. Although this approach facilitated the most robust anatomical regeneration of the CST after a spinal cord crush (SCC) so far[10], it fails to improve functional motor recovery[11].

In the optic nerve, the activation of the Janus kinase/signal transducer and activator of transcription 3 (JAK/STAT3) pathway stimulates the regeneration of CNS axons[12,13]. JAK/STAT3 activation is achieved via the delivery of IL-6- type cytokines such as CNTF, LIF, IL-6, and/or the genetic depletion of the intrinsic STAT3 feedback inhibitor suppressor of cytokine signaling 3 (SOCS3)[12,14–17]. However, the low and restricted expression of the cytokine-specific α-receptor subunits in CNS neurons required for signaling induction limits the pro-regenerative effects of native cytokines. For this reason, a gene therapeutic approach was recently developed using the designer cytokine hyper-interleukin-6 (hIL-6), which consists of the bioactive part of the IL-6 protein covalently linked to the soluble IL-6 receptor α subunit[18]. In contrast to natural cytokines, hIL-6 can directly bind to the signal-transducing receptor subunit glycoprotein 130 (GP130) abundantly expressed by almost all neurons[19,20]. Therefore, hIL-6 is as potent as CNTF but activates cytokine-dependent signaling more effectively[18]. In the visual system, virus-assisted gene therapy with hIL-6, even when applied only once postinjury, induces more robust optic nerve regeneration than a pre-injury induced PTEN knockout (PTEN$^{-/-}$)[18]. Hence, this treatment is the most effective approach to stimulate optic nerve regeneration when applied after injury.

The current study analyzed the effect of cortically applied AAV-hIL-6 alone or in combination with PTEN$^{-/-}$ on functional recovery after severe SCC. A one-time unilateral injection of AAV-hIL-6 applied after SCC into the sensorimotor cortex promoted regeneration of CST-axons stronger than PTEN$^{-/-}$, and, additionally, serotonergic fibers of the raphespinal tract, which enabled locomotion recovery of both hindlimbs. Moreover, we demonstrate that cortical motoneurons project collaterals to serotonergic raphe neurons deep in the brain stem, allowing the release of hIL-6 to induce regenerative stimulation of serotonergic neurons. Thus, transneuronal stimulation of neurons located deep in the brain stem using highly potent molecules might be a promising strategy to achieve functional repair in the injured or diseased human CNS.

## Results

### Hyper-IL-6 and PTEN knockout activate different signaling pathways. To investigate the impact of hIL-6 and PTEN$^{-/-}$ on

regeneration associated signaling pathways in motoneurons, we applied either AAV2-hIL-6 (GFP coexpression), AAV2-Cre (GFP coexpression), or AAV2-GFP into the sensorimotor cortex of adult PTEN-floxed mice. Viral transduction affected mainly layer V neurons in the primary motor cortex, adjacent to the injection sites (Fig. 1a–c and Fig. S1a–e). Moreover, only AAV-hIL-6 transduced cells showed the hIL-6 protein (Fig. S1f, g). Western blot analysis verified these results, detecting hIL-6 in respective tissues of AAV-hIL-6 treated animals only (Figs. S1h, i, S15d, e, S16n, o). In contrast to AAV2-Cre (PTEN$^{-/-}$) and AAV2-GFP, AAV2-hIL-6 application induced strong STAT3-phosphorylation as shown in sections of cortical tissue and western blot lysates (Fig. 1b–g, k, m, n, o and Fig. S1j, k). These signals were seen in all GFP-positive hIL-6-transduced neurons and adjacent cells, indicating the paracrine effects of released hIL-6. STAT3 activation was already at its maximum 1 week after injection and remained stable over at least 8 weeks (Fig. 1n, o, Fig. S14a-d, f-h; Fig. S16a), while total STAT3 protein expression was not significantly altered (Fig. 1n, p and Figs. S14f, i-m, Fig. S16b, d). In contrast to PTEN$^{-/-}$, which induced robust phosphorylation of AKT (pAKT) and S6 (pS6), AAV2-hIL-6 had little impact on the phosphorylation of these proteins (Fig. 1h–l, n, q, r, t and Figs. S14n, S16c, f, g). PTEN$^{-/-}$-induced phosphorylation of S6 was restricted to GFP-positive (transduced) neurons only (Fig. 1k, l). Neither hIL-6 nor PTEN$^{-/-}$ influenced phosphorylation of ERK1/2 (Fig. 1r, s; Fig. S14 m, o-p and Fig. S16e).

### Hyper-IL-6 promotes CST regeneration. Next, we tested whether AAV2-hIL-6 application, PTEN$^{-/-}$, or their combination affect corticospinal tract (CST) regeneration following complete spinal cord crush (SCC). As AAV2 reaches higher neuronal transduction rates in newborn animals[10,11], and to keep the methodology comparable to these previous studies, PTEN-floxed mice received injections of either AAV2-Cre (PTEN$^{-/-}$) or AAV2-GFP (PTEN$^{+/+}$) into the left sensorimotor cortex at postnatal day 1 (P1). After 7 weeks, mice were subjected to SCC (Fig. 2a). Each received a second injection of either AAV2-hIL-6 or AAV2-GFP into the left sensorimotor cortex immediately afterward (Fig. 2a), resulting in four experimental groups: (a) Control animals that had received AAV2-GFP injections twice (PTEN$^{+/+}$), (b) PTEN-floxed mice that were treated with AAV2-Cre and later with AAV2-GFP (PTEN$^{-/-}$), (c) mice that received AAV2-GFP and later, after SCC, AAV2-hIL-6 (hIL-6) and (d) mice that were treated with AAV2-Cre first and, after SCC, with AAV2-hIL-6 (PTEN$^{-/-}$/hIL-6). Axonal biotinylated dextran amine (BDA)-tracing of the right CST (Fig. S2j) was performed 6 weeks after SCC, followed by the collection of the brains and spinal cords for imaging 2 weeks after that (Fig. 2a). Neither PTEN$^{-/-}$, hIL-6 expression nor their combination affected the total number of BDA-positive CST axons in the medullary pyramid (Fig. S2a, b) or ipsilateral CST-axonal sprouting in the thoracic spinal cord rostral to the lesion site compared to controls (Fig. S2c–i). Likewise, the lesion size and width were constant among all groups (Fig. S2k–m). Moreover, cortical AAV2-hIL-6 treatment did not activate or attract any macrophages or microglia in the spinal cords of uninjured mice (Fig. S3a). It also did not increase the number of CD11b-positive cells in the lesion site (Fig. S3b–e), either. However, axonal dieback of CST-fibers above the injury site, typically seen in control mice, was significantly reduced by PTEN$^{-/-}$ and reduced slightly further by AAV2-hIL-6 treatment (Fig. 2b, c and Fig. S2n–r). The combination of PTEN$^{-/-}$ + AAV2-hIL-6 showed a slight, but significant additional effect (Fig. 2b, c).

We then analyzed fiber regeneration beyond the crush site. Contrary to controls (Fig. 2d, e, i and Fig. S4a), PTEN$^{-/-}$ enabled

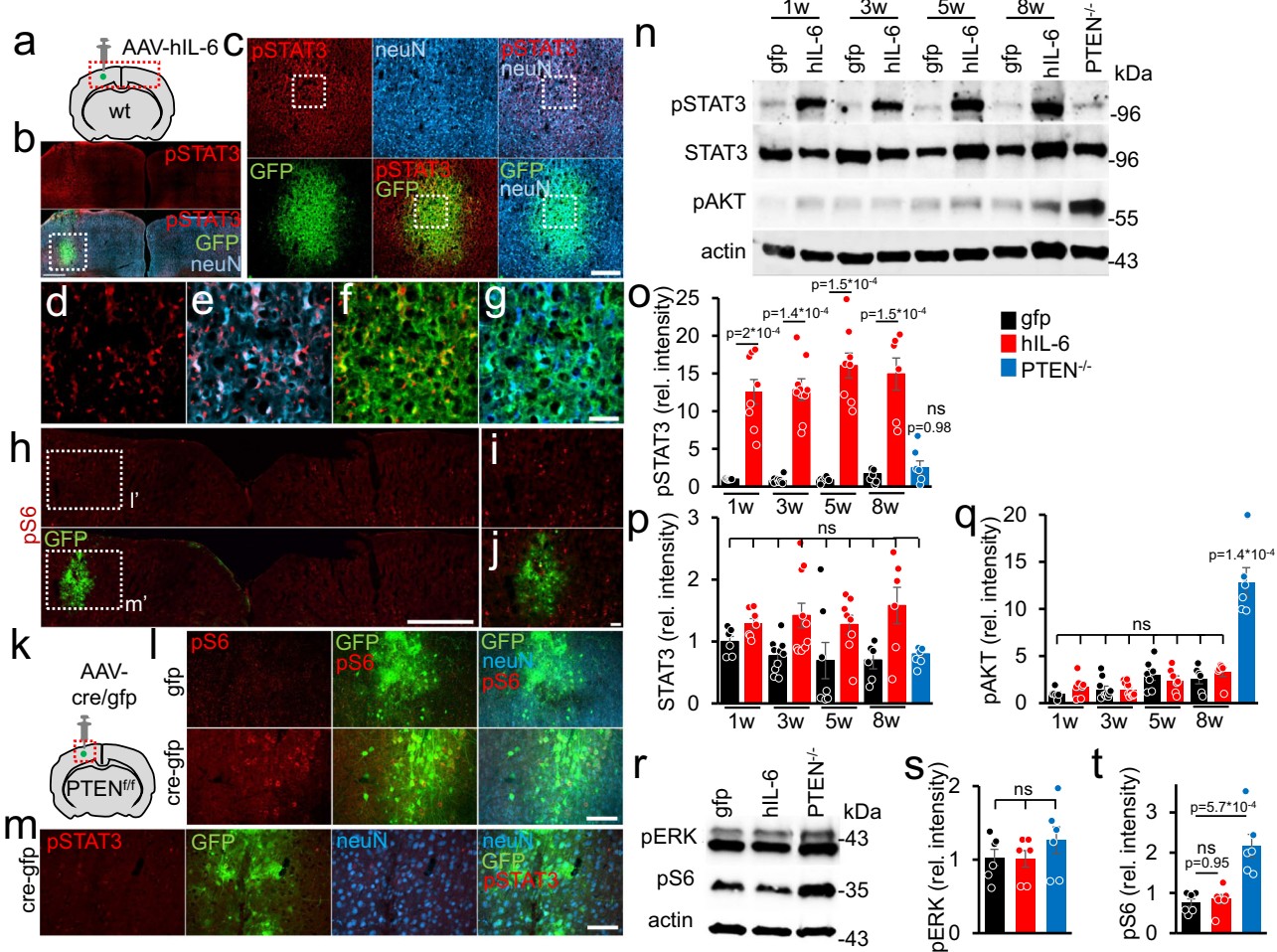

**Fig. 1 Activation of signaling pathways by hIL-6 or PTEN$^{-/-}$. a** Schematic drawing illustrating the location of the AAV-injection sites (dashed box) shown in (**b**). **b** Coronal section of the sensorimotor cortex from a wild-type mouse 3 weeks after intracortical injection of AAV2-hIL-6 into the left hemisphere. The section was immunohistochemically stained for phosphorylated STAT3 (pSTAT3, red) and neuN (blue). GFP (green) was co-expressed by the AAV2-hIL-6. Scale bar: 500 μm. **c** Higher magnification of dotted box shown in (**b**). Scale bar: 200 μm. **d–g** Higher magnification of dotted boxes in (**c**). Scale bar: 50 μm. **h–j** Immunohistochemical staining against phosphorylated ribosomal protein S6 (pS6, red) of sections as described in (**a**). Dashed boxes are presented at higher magnification in (**i, j**). Scale bars: 50 μm (**i, j**); 500 μm (**h**). **k** Schematic drawing illustrating the injection site and location of images shown in (**l, m**). **l, m** Cortical sections of PTEN$^{f/f}$ mice 3 weeks after AAV2-Cre-GFP (Cre-gfp), or AAV2-GFP (gfp) injections, stained for pS6 (**l**, red), or pSTAT3 (**m**, red). Scale bar: 50 μm. **n** Western blot analysis: lysates prepared from the sensorimotor cortex of PTEN$^{f/f}$ mice 1, 3, 5, or 8 weeks (w) after intracortical injection of either AAV2-hIL-6 (hIL-6), AAV2-GFP (gfp) or 5 weeks after AAV2-Cre application leading to PTEN$^{-/-}$. AAV2-hIL-6 induced STAT3 phosphorylation (pSTAT3) at all tested time points, while PTEN$^{-/-}$ only caused AKT phosphorylation. Total STAT3 protein remained mostly unaltered. Beta-actin served as a loading control. **o–q** Densitometric quantifications of western blots depicted in (**n**). Values represent means ± SEM of samples from 6 to 10 animals per group (gfp 1w, n = 6; hil-6 1w, n = 8; gfp 3w, n = 10; hIL-6 3w n = 10; gfp 5w, n = 7; hIL-6 5w, n = 8; gfp 8w, n = 6; hIL-6 8w, n = 6; PTEN$^{-/-}$, n = 6). **r** Western blot analysis of cortical lysates: Phosphorylation of ERK1/2 (pERK) was not altered by AAV2-GFP-, AAV2-hIL-6, or AAV2-Cre (PTEN$^{-/-}$) 5 weeks after intracortical application. Only PTEN$^{-/-}$ induced significant S6-phosphorylation (pS6). **s/t** Densitometric quantification of western blots depicted in (**r**). Values represent means ± SEM of 6 independent cortical lysates (n = 6) per group. Representative immunohistochemical stainings shown in (**b–j, l, m**) were repeated four times with individual biological samples with similar results. Significances of intergroup differences in (**o–q, s, t**) were evaluated using a one-way analysis of variance (ANOVA) followed by Tukey post hoc test. Statistical significance is indicated by p-values. ns = non-significant. Dots in **o–q, s, t** represent values of single samples. Source data are provided as a Source data file.

the regeneration of some CST axons, most of which did, however, not exceed distances greater than 1.5 mm (Fig. 2d, f, i and Fig. S4a). In contrast, AAV2-hIL-6 treatment resulted in more robust CST regeneration with the longest axon reaching up to 6 mm (Fig. 2d, g, i and Fig. S4a). This effect was slightly increased further by the combination of PTEN$^{-/-}$ and AAV2-hIL-6, with the longest axons reaching >7 mm (Fig. 2d, h, i and Fig. S4a). No BDA labeled axons were detected >11 mm past the lesion site, thereby excluding spared axons and verifying the completeness of axonal injury in all animals (Fig. S5a–d).

**Hyper-IL-6 promotes functional recovery**. Before tissue isolation and analysis, hindlimb movement had been analyzed in all four experimental groups using open-field locomotion tests according to the Basso Mouse Scale (BMS) over the postinjury period of 8 weeks[21]. Consistent with previous reports[21–23], the BMS score dropped down to 0 in all animals 1 day after injury, also indicating the completeness of the SCC (Fig. 3a, b and Fig. S6a–e). Over time, control animals developed only active ankle movements, including spasms (Supplementary video 1)[21,22], resulting in an average final score of 2 (Fig. 3a, b and Fig. S6a). Despite the positive effect on

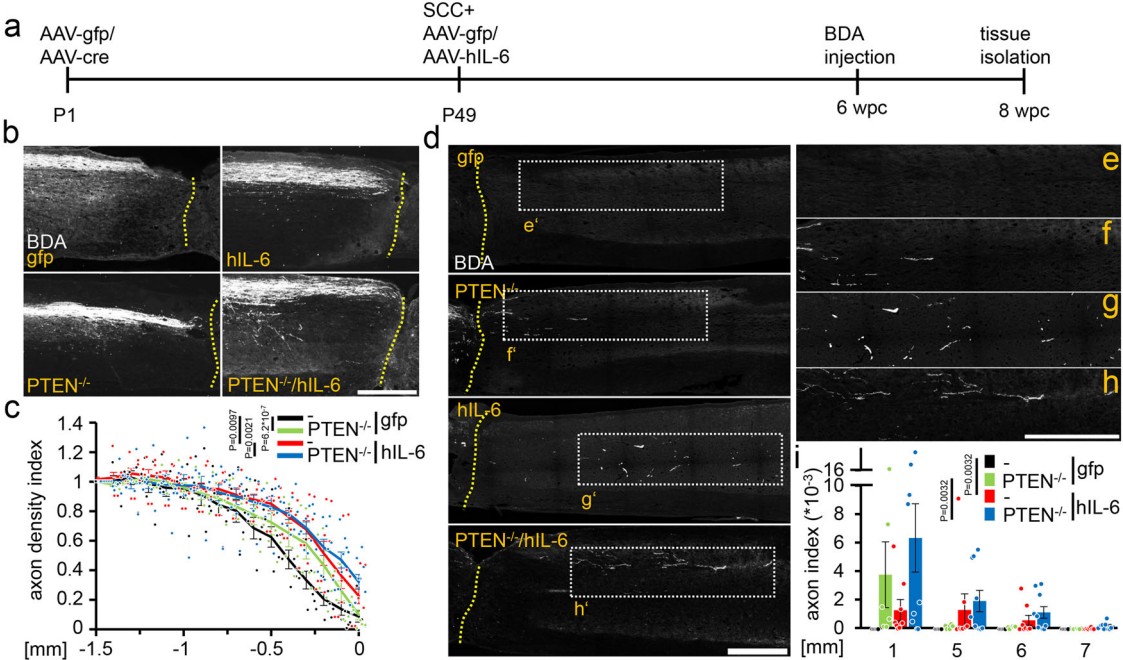

**Fig. 2 Hyper-IL-6 promotes CST axon regeneration after severe spinal cord crush. a** Timeline of surgical interventions for experiments presented in (**b**–**i**) and Figs. 3, 4. PTEN$^{f/f}$ mice received unilateral injections of AAV2-GFP (PTEN$^{+/+}$) or AAV2-Cre (PTEN$^{-/-}$) at P1. After 7 weeks, PTEN$^{+/+}$ and PTEN$^{-/-}$ mice were then subjected to spinal cord crush (SCC) (T8) and subsequently received additional unilateral intracortical injections of either AAV2-hIL-6 (hIL-6) or AAV2-GFP (gfp). **b** Sagittal thoracic spinal cord sections of four differently treated groups showing BDA-labeled, retracting axons (white) of the right main CST rostral to the lesion site (dotted line). Tissues were isolated 8 weeks after the SCC. Scale bar: 500 μm. **c** Axon density index of CST fibers analyzing all spinal cord sections with the main CST of the animals described in (**a**, **b**). Values at distances from 1.5 mm rostral to the lesion site (−1.5) up to the proximal lesion border (0) were determined. Values represent means ± SEM of 6–10 animals per group (PTEN$^{+/+}$/gfp, n = 7; PTEN$^{+/+}$/hIL-6, n = 9; PTEN$^{-/-}$/gfp, n = 6; PTEN$^{-/-}$/hIL-6, n = 10). **d** Representative images of BDA-labeled sagittal spinal cord sections from PTEN-floxed mice (PTEN$^{f/f}$) after treatments, as described in (**a**, **b**). Tissues were isolated after 8 weeks. BDA-labeled (white) regenerating axons of the central CST beyond the lesion site (dotted line) were seen only after PTEN$^{-/-}$, hIL-6, or hIL-6/PTEN$^{-/-}$-treatments. Scale bar: 500 μm. **e**–**h** Higher magnification of regenerating axons in dotted boxes as indicated in (**d**). Scale bar: 500 μm. **i** Quantification of regenerating CST axons at the indicated distances caudal to the lesion site. Axon numbers were divided by the total number of BDA-labeled CST fibers in the medulla (axon index) from animals as described in (**a**, **d**). Values represent the mean ± SEM of 6–10 animals per group (PTEN$^{+/+}$/gfp, n = 7; PTEN$^{+/+}$/hIL-6, n = 9; PTEN$^{-/-}$/gfp, n = 6; PTEN$^{-/-}$/hIL-6, n = 10). Significances of intergroup differences in (**c**, **i**) were evaluated using a three-way analysis of variance (ANOVA) with Holm Sidak post hoc tests. Statistical significance is indicated by p-values; ns = non-significant. Dots in **c** and **i** represent values of single animals. Source data are provided as a Source data file.

CST-regeneration (Fig. 2d, f, i), PTEN$^{-/-}$ alone did not significantly improve the BMS score compared to AAV2-GFP treated controls (Fig. 3a, b and Fig. S6c; Supplementary video 2). However, AAV2-hIL-6 treatment increased the BMS score to ~4 by restoring plantar stepping with full hindlimb weight support, followed by lift-off, forward limb advancement, and reestablishment of weight support at initial contact in most of the animals (Fig. 3a, b and Fig. S6b; Supplementary video 3). Combinatorial treatment (PTEN$^{-/-}$ + AAV2-hIL-6) slightly enhanced this effect further (Fig. 3a, b and Fig. S6d, Supplementary video 4). These effects were best seen in the BMS subscore (Fig. 3e). One animal of this group reached a coordinated movement of fore- and hindlimbs (score: 7) (Fig. 3a, b and Fig. S6d). Interestingly, despite only administering a unilateral AAV2-hIL-6 injection into the left sensorimotor cortex, both hindlimbs showed similar recovery (Fig. 3a, c, d). Moreover, a bilateral application into the left and right side had no additional effect (Fig. 3f and Fig. S6e). Thus, in contrast to PTEN$^{-/-}$, a single unilateral postinjury application of AAV2-hIL-6 into the sensorimotor cortex enabled both hindlimbs' locomotion after severe SCC. The effect on functional recovery was also verified using an automated catwalk gait analysis system. To this end, we generated another cohort of injured mice with either unilateral AAV2-hIL-6 or AA2V-GFP application. While all AAV2-GFP-treated mice failed to show any hind paw placement, functional restoration after AAV2-hIL-6 treatment

allowed for the quantification of several parameters, such as the area of hind paw footprints during contact with the glass plate, which was similar to uninjured controls (Fig. S7b–f). Also, stride length (Fig. S7g, h), the base of support (BOS) (Fig. S7i), and the regularity index were measured and reached about 40-60% of the pre-injury values. The latter indicates the degree of coordination between hind and forelimbs (Fig. S7j and Supplementary video 5). Thus, BMS (Fig. S7a) and Catwalk evaluation showed similar results concerning the ranking of individual animals indicated by their linear correlation (Fig. S7k).

**Cortical AAV2-hIL-6 delivery promotes RpST regeneration.** The lack of functional recovery in PTEN$^{-/-}$ mice suggested that improved CST regeneration was not the leading cause for the AAV2-hIL-6 mediated functional recovery. As descending serotonergic (5-HT-positive) axons of the raphespinal tract (RpST) are reportedly relevant for locomotor recovery[24–26], we stained and analyzed these axons in sagittal spinal cord sections of the same animals used in the previous experiments (Figs. 2, 3). Control (AAV2-GFP) and PTEN$^{-/-}$ mice only revealed sprouting of 5-HT-positive axons over distances of less than 1 mm beyond the crush site (Fig. 4a–e, j and Fig. S4c). Remarkably, AAV2-hIL-6 treated mice showed more, and longer-distance regeneration of serotonergic fibers than controls or PTEN$^{-/-}$

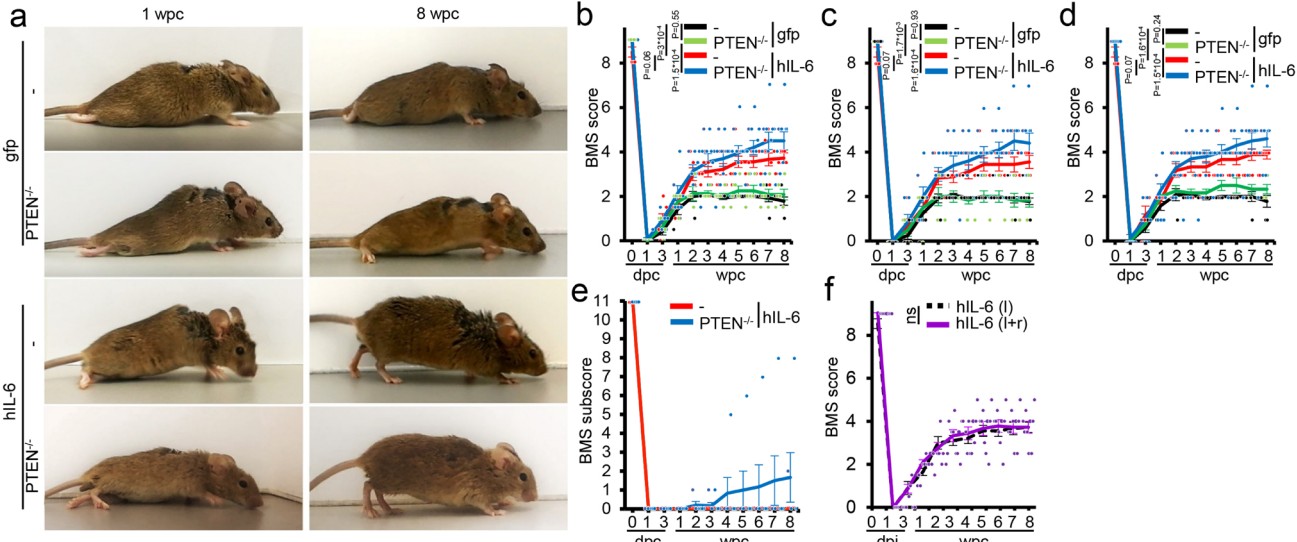

**Fig. 3 Hyper-IL-6 enables functional recovery after SCC. a** Representative pictures are showing open field hindlimb movement of mice at 1 and 8 weeks after spinal cord crush (wpc) and treatment, as described in Fig. 2. **b–d** BMS score of animals as described in (**a**). Scores were evaluated at 0, 1, 3, 7 days post crush (dpc) and then weekly over 8 weeks after spinal cord injury. Values represent means ± SEM of 6–10 animals per group (PTEN$^{+/+}$/gfp, $n = 7$; PTEN$^{+/+}$/hIL-6, $n = 9$; PTEN$^{-/-}$/gfp, $n = 6$; PTEN$^{-/-}$/hIL-6, $n = 10$), showing either the average score of the left and right hind paw (**b**) or left (**c**) and right (**d**) side separately. **e** BMS subscore of hIL-6 treated PTEN$^{+/+}$(−) and PTEN$^{-/-}$ mice as described in (**a**). **f** Average BMS score of left and right hind paws from mice after SCC and bilateral (left and right hemisphere (l + r)) intracortical injection of AAV2-hIL-6 compared to animals that had received a unilateral injection into the left hemisphere (l) only as presented in (**b**). Values represent the mean ± SEM of 9 animals per group (l, $n = 9$; l + r, $n = 9$). Significances of intergroup differences were evaluated using a two-way analysis of variance (ANOVA) with a Tukey post hoc test (**b–d**) or two-sided student's *t*-test (**f**). *P*-values indicate statistical significance; ns = non-significant. Source data are provided as a Source data file.

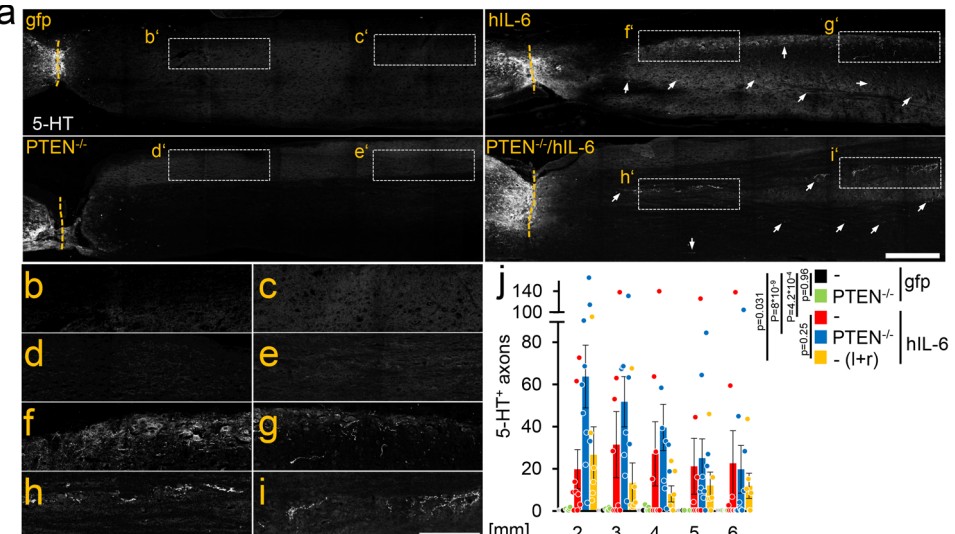

**Fig. 4 hIL-6 promotes axon regeneration of serotonergic fibers. a** Sagittal thoracic spinal cord sections isolated from PTEN$^{+/+}$, or PTEN$^{-/-}$ animals 8 weeks after spinal cord crush (SCC) and unilateral injection of AAV2-hIL-6 (hIL-6) or AAV2-GFP (gfp) (see Fig. 2). Raphe spinal tract (RpST) axons were stained using an anti-serotonin antibody (5-HT, white). Only AAV2-hIL-6-treated mice with or without additional PTEN$^{-/-}$ showed significant regeneration of serotonergic axons beyond the lesion site (dashed line). As typical for regenerating RpST axons they were located over the whole dorsoventral width of the spinal cord. Examples are indicated by dashed boxes and white arrows. Scale bar: 500 μm. **b–i** Higher magnification of dashed boxes as indicated in (**a**). Scale bar: 250 μm. **j** Quantification of regenerating 5-HT-positive axons as described in (**a**) at indicated distances beyond the lesion. Values represent the mean ± SEM of 6–10 animals per group (PTEN$^{+/+}$/gfp, $n = 7$; PTEN$^{+/+}$/hIL-6, $n = 9$; PTEN$^{-/-}$/gfp, $n = 6$; PTEN$^{-/-}$/hIL-6, $n = 10$; PTEN$^{+/+}$/hIL-6 (l + r); $n = 6$). Significances of intergroup differences were evaluated using a two-way analysis of variance (ANOVA) with a Holm Sidak post hoc test. Dots in **j** represent values of single animals. *P*-values indicate statistical significance; ns = non-significant. Source data are provided as a Source data file.

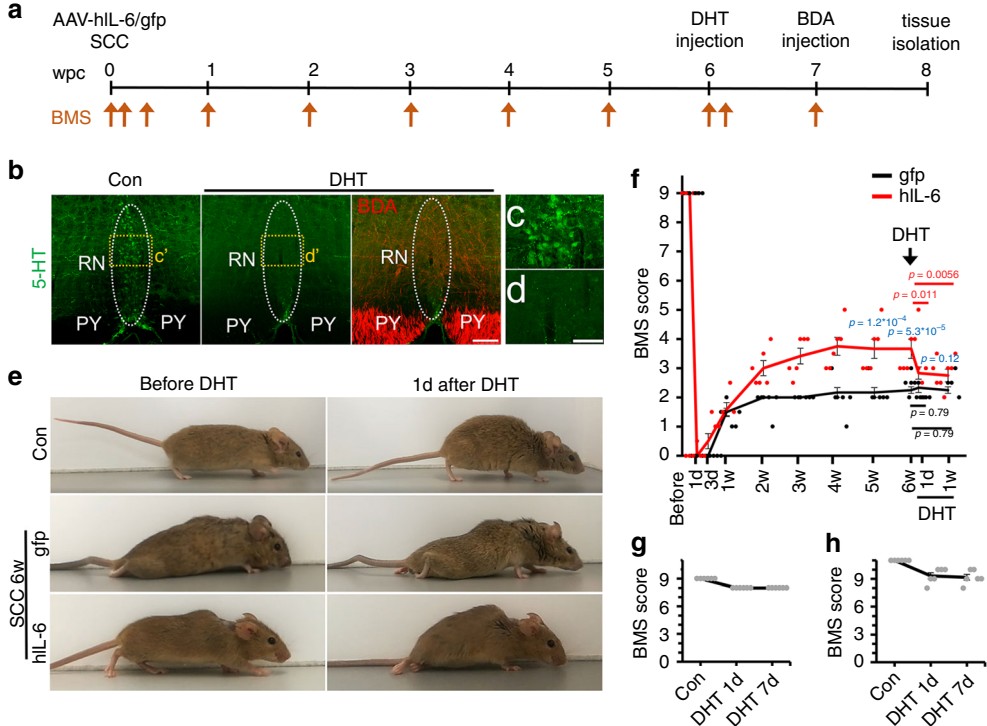

**Fig. 5 Regeneration of serotonergic axons is essential for functional recovery. a** Timeline of experiments shown in (**b–f**). Adult mice were subjected to SCC and received bilateral intracortical AAV2-GFP (gfp) or AAV2-hIL-6 (hIL-6) injections. They were then scored according to the Basso Mouse Scale (BMS) at the indicated time points (arrows) over 8 weeks post crush (wpc). Six weeks after SCC, the serotonin neurotoxin 5,7-dihydroxytryptamine (DHT) was injected intracerebroventricularly into both hemispheres. One week before tissue isolation, BDA was intracortically injected to trace CST axons. **b** Maximum intensity projection of confocal scans through 50 μm of cleared brain stem tissue from mice that received hIL-6 and DHT treatment as described in (**a**) compared to a control (con) without DHT treatment. Serotonergic neurons of raphe nuclei (RN, dotted line) were visualized by 5-HT staining (green). BDA-labeled CST axons in the pyramid (PY) are shown in red. DHT treatment eradicated almost all 5-HT-positive neurons without affecting corticospinal neurons, indicated by intact BDA labeled axons. Scale bar: 100 μm. **c, d** Higher magnifications of dashed boxes (**c'** and **d'**) in (**b**). Scale bar: 50 μm. **e** Representative images of AAV2-GFP (gfp) or AAV2-hIL-6 (hIL-6)-treated mice 6 weeks post crush (wpc), or untreated mice (con) before and 1 day after DHT injection. **f** BMS score of animals treated as described in (**a**) at indicated time points after spinal cord crush and DHT treatment. Values represent means ± SEM of 6 animals per group ($n = 6$), showing the average score of left and right hind paws. **g, h** BMS scores (**g**) and subscores (**h**) of untreated mice before (con) and 1 day (DHT 1d), or 1 week (DHT 7d) after DHT injection. Values represent means ± SEM of 6 animals showing the average score of left and right hind paws. Significances of intergroup differences were evaluated using a two-way analysis of variance (ANOVA) with a Holm Sidak post hoc test. Significance indicated by p-values within the hIL-6 group in red, within the GFP group in black, or between hIL-6 and GFP group in blue. Source data are provided as a Source data file.

(Fig. 4a, f, g, j and Fig. S4c), with the longest axons reaching distances of >7 mm. Combinatorial treatment of AAV2-hIL-6 and PTEN$^{−/−}$ had no additional effect (Fig. 4a, h, i, j and Fig. S4c). Furthermore, as measured during functional recovery analysis (Fig. 3f), bilateral AAV2-hIL-6 treatment did not increase RpST regeneration further compared to a unilateral application of the virus (Fig. 4j and Fig. S4c), and no 5-HT-positive axons were detected >11 mm beyond the injury site in any of these mice verifying the absence of spared axons.

**AAV2-hIL-6-mediated recovery depends on the regeneration of serotonergic neurons.** To investigate the relevance of RpST regeneration for functional recovery, adult mice of the same genetic background were subjected to SCC and received bilateral intracortical injections of either AAV2-hIL-6 or AAV2-GFP directly afterward (Fig. 5a). At week 6, when functional recovery in AAV2-hIL-6 treated mice had already reached maximal levels, the neurotoxin 5,7-dihydroxytryptamine (DHT) was intracerebroventricularly injected to selectively kill serotonergic neurons (Fig. 5a–d)[24,27,28]. The toxin almost completely abolished 5-HT positive neurons/axons in the medulla/spinal cord as soon as 1 day after injection (Fig. S8a–j). As described previously[27,29,30],

other neurons remained unaffected, indicated by the successful BDA tracing of cortical motor neurons (Fig. 5b), presence of other neuN positive neurons next to depleted Raphe neurons in the medulla (Fig. S8k), and the almost normal open field movement in uninjured mice evaluated 1 week after DHT application (Fig. 5e, g, h). However, in injured animals, 1 day after DHT application, the BMS of AAV2-hIL-6-treated mice dropped down to similar levels as in AAV-GFP treated control animals, which themselves remained unaffected by the DHT treatment and thereby verified the neurotoxin effect's specificity (Fig. 5e, f and Supplementary video 6). Thus, AAV2-hIL-6-mediated locomotory recovery depended on RpST regeneration, even though AAV2-hIL-6 had not been applied to the brain stem where the serotonergic fibers originated.

Since genetic backgrounds could potentially affect regeneration, we also tested the postinjury-applied AAV2-hIL-6 treatment in non-transgenic BL6 mice, whose low potential for functional recovery has been previously documented[21]. Eight weeks after SCC and unilateral intracortical AAV2-hIL-6 injection, BL-6 mice also showed reduced axonal dieback, but less CST-regeneration caudal to the lesion site than accordingly treated PTEN-floxed OLA mice (Figs. S9a, b, S4b). Nevertheless, the

improvement in RpST axon regeneration and functional recovery was very similar in both mouse strains (Figs. S9c–e, S4d; Supplementary video 7), supporting a correlation between anatomical RpST regeneration and functional recovery.

**AAV2-hIL-6 confers transneuronal stimulation of serotonergic fiber regeneration.** To understand the mechanism underlying the bilateral, RpST-dependent functional recovery by a one-time unilateral AAV2-hIL-6 virus application into the sensorimotor cortex, we tested the hypothesis that transduced cortical motoneurons project axons to the raphe nuclei in the brain stem and that the release of hIL-6-protein stimulates regeneration of serotonergic neurons trans-synaptically. Starting in vitro, we used axon isolation devices to separate somata and axons of cultured sensory DRG neurons[31] (Fig. S10a) and transduced these neurons by adding baculoviruses (BV)[18,32] for either hIL-6 or GFP expression into the somal chamber. Western blot analysis detected hIL-6 protein in the medium of the axon chamber of hIL-6 transduced neurons only, verifying its release (Figs. S10c, S16p, q). Moreover, synaptotagmin-positive vesicles containing hIL-6 were found in the axons and their tips (Fig. S10d). To test the activity of the released hIL-6, the medium of the axonal chamber was used to prepare dissociated cultures of adult retinal ganglion cells (RGC) (Fig. S10b). Consistent with previous findings[18], the conditioned medium from hIL-6-transduced sensory axons induced STAT3 phosphorylation (Fig. S10e) and increased neurite outgrowth of RGCs twofold compared to the medium from GFP controls (Fig. S10f, g). Thus, hIL-6 is transported in axons, released at the terminals, and remains active to stimulate axon regeneration.

To investigate whether transduced hIL-6 can also stimulate neurons trans-synaptically in vivo, we injected either AAV2-hIL-6 or AAV2-GFP into the left eye of mice to transduce RGCs. Three weeks later, we tested for STAT3 phosphorylation in their brain targets, the lateral geniculate nucleus (LGN) and suprachiasmatic nucleus (SCN) (Fig. S10h). In rodents, ~95% of the optic nerve axons cross in the optic chiasm to the contralateral side. Consistent with a synaptic release of hIL-6, we found many pSTAT3-positive cellular nuclei near GFP-positive axon terminals of AAV2-hIL-6-transduced RGCs in the LGN and SCN of the right hemisphere, with much less found in the respective left hemispheres and no signals at all in AAV2-GFP treated controls (Fig. S10i, j). The hIL-6 expression in RGCs did not affect axons' integrity as their number, determined after neurofilament staining in the optic nerve and optic tract cross-sections, was not reduced compared to untreated controls, excluding any uncontrolled or widespread release of the protein.

We then investigated whether raphe neurons in the brain stem, which express the hIL-6 receptor GP130[33,34] receive synaptic input from transduced cortical motoneurons. To this end, we analyzed 5-HT stained brain stem tissue from mice 8 weeks after SCC and cortical AAV2-hIL-6 treatment as described above (Fig. 2a). GFP-positive sprouts of cortical motoneurons were located near 5-HT-positive raphe neurons in cross-sections of the medulla (Fig. S11a–d). The GFP signal and hIL-6 in the pyramidal axons were detected via high-resolution transversal scans after tissue clearing (Fig. S1l–n). Western blot analyses using brain stem lysates isolated 3 weeks after intracortical viral application showed pSTAT3 signals only in samples from AAV2-hIL-6-, but not AAV2-GFP-treated animals (Fig. 6a–c and Fig. S15a-c, S16h), while total STAT3 protein expression or phosphorylation of either AKT or S6 remained unchanged (Fig. 6b, c and Figs. S15, S16h–m). Additionally, transverse scans through cleared brain stem tissue (Fig. S12a, b, Supplementary video 8) from hIL-6 treated animals, and immunostaining of

coronal sections showed, on average, 46% ± 4.46 SEM of the serotonergic neurons were pSTAT3/5-HT double-positive. These neurons were close to GFP-positive axons in the ipsi- and contralateral sides of the raphe nuclei in the medial column of the medulla. 50.63 ± 4.4% of double-positive neurons were identified in the ipsilateral side of the raphe nuclei and 49.37 ± 4.4% on the contralateral side. These findings were not observed in AAV-GFP treated controls (Fig. 6d–u). Moreover, intracortical AAV2-hIL-6 treatment also activated STAT3 in neurons of the red nucleus, a known target of cortical motoneurons (Fig. S12c–e)[35], thereby corroborating the transneuronal delivery of hIL-6.

To verify that cortical motoneurons were synaptically connected with raphe neurons, we used AAV1, which, in contrast to AAV2, can trans-synaptically transduce supraspinal target neurons[36]. We injected AAV1-Cre into the left sensorimotor cortex of Rosa26-tdTomato (RFP) reporter mice and isolated their brain stem tissue after 2 weeks. RFP-positive (transduced) serotonergic neurons were detected in both sides of the nucleus raphe pallidus (NRPa) and the nucleus raphe magnus (NRM) (Fig. 7a–f) along their entire rostrocaudal length. Additionally, we performed a complementary experiment by injecting AAV1 directly into the raphe nuclei. As AAV1 is also retrogradely transported[37,38], we found transduced neurons in the motor cortex layer V, confirming the synaptic connection to cortical neurons (Fig. 7g–i). Thus, cortically applied AAV2-hIL-6 transneuronally activates regenerative signaling pathways in ipsi- and contralateral raphe neurons by releasing the protein at the axon terminals.

## Discussion

The current study shows significant functional recovery after a complete SCI in adult mammals. This recovery was achieved by the one-time unilateral application of AAV2-hIL-6 into the sensorimotor cortex. While pre-injury PTEN$^{-/-}$ in cortical neurons failed to facilitate functional recovery, postinjury AAV2-hIL-6 application enabled locomotor recovery of both hindlimbs. Moreover, this treament promoted longer CST axon growth than PTEN$^{-/-}$ and, additionally, regeneration of serotonergic axons in the RpST. We also provide evidence that cortical motoneurons directly innervate raphe neurons, allowing the delivery of highly potent hIL-6 to stimulate the regeneration of these serotonergic brain stem neurons. Thus, the transneuronal application of highly active molecules has proven to be a powerful approach to activate regenerative signaling, particularly in neurons located in brain regions that are challenging to access but relevant for functional recovery after spinal cord injury.

Incomplete, less severe hemisection- or contusion-injury models permit spontaneous functional recovery with BMS scores ≥6[11,21,23,39,40]. In contrast, the SCC, just as in a complete transection, severs all axonal connections between the proximal and distal portions of the spinal cord, allowing only some active flexion movements of hindlimbs (BMS score of ≤2)[10,21–23,41]. Therefore, achieving functional recovery in this injury model is challenging, and most previous treatment strategies tested did not reach significant effects beyond reducing the dieback of CST-axons[42–44]. Only PTEN$^{-/-}$ in cortical motoneurons enabled the regeneration of CST-axons and their synapse formation with interneurons, but still failed to restore locomotion[10,11,41,45,46]. Therefore, we used the PTEN$^{-/-}$ model as a reference to validate the effects of AAV2-hIL-6. While cortical PTEN$^{-/-}$ showed regenerated axons only at short distances of up to 1 mm beyond the injury site, distances were markedly longer in hIL-6 treated animals. Moreover, only AAV2-hIL-6 treatment significantly improved serotonergic axon growth in the RpST and enabled locomotion recovery of both hindlimbs. These effects

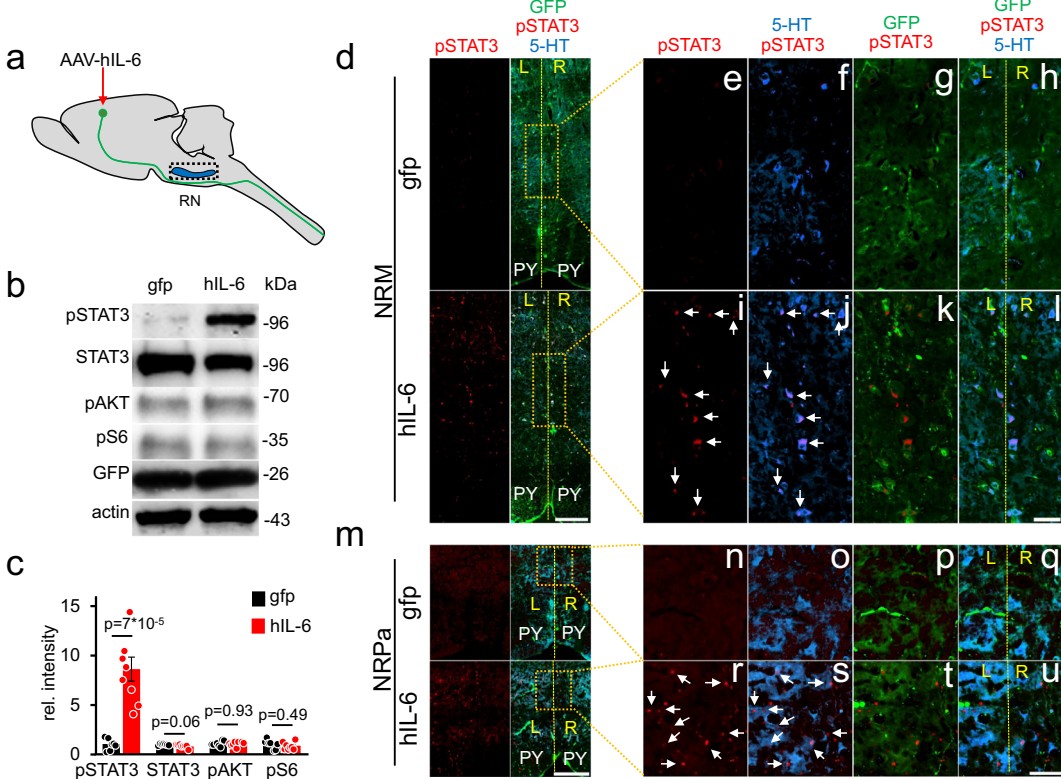

**Fig. 6 Hyper-IL-6 transneuronally stimulates neurons of raphe and red nuclei. a** Schematic of cortical AAV2-hIL-6 application and isolated brain stem tissue (dashed box) containing the raphe nuclei (RN) for western blot analysis. **b** Western blots: GFP, total (STAT3) and phosphorylated STAT3 (pSTAT3), phospho-AKT (pAKT), and phospho-S6 (pS6) were analyzed in lysates of the brain stem with raphe nuclei isolated 3 weeks after intracortical injection of either AAV2-GFP (gfp) or AAV2-hIL-6 (hIL-6). GFP signals in lysates verified similar amounts of transduced collateral axons that projected to the brain stem. Beta-actin served as a loading control. **c** Densitometric quantification of western blots from 7 to 8 animals per group (gfp, $n = 7$; hIL-6, $n = 8$) as described in (**a**). Data represent means ± SEM. **d**, **m** Immunostaining of brain stem sections containing the nucleus raphe magnus (NRM, **d**), or nucleus raphe pallidus (NRPa, **m**) of AAV2-GFP or AAV2-hIL-6 treated mice as described in Fig. 2a. Sections were stained for pSTAT3 (red), GFP (green), and serotonin (5-HT, blue) to identify raphe neurons. The dashed yellow line indicates the midline. We observed a similar amount of pSTAT3 positive serotonergic neurons in the left and right hemisphere of the raphe nuclei from AAV2-hIL6 treated mice ($n = 6$). Scale bar: 200 μm. **e**, **f**, **n–u** Higher magnifications of the dotted box as indicated in (**d**) (**e–l**) or (**m**) (**n–u**). Significances of intergroup differences in (**c**) were evaluated using the two-sided student's *t*-test and indicated by *p*-values. Dots in **c** represent values of samples from single animals. Source data are provided as a Source data file.

were achieved in mice with different genetic backgrounds by a one-time, unilateral application of AAV2-hIL-6 after the SCC, making this gene therapeutic approach a potential strategy to facilitate spinal cord repair.

PTEN$^{-/-}$ and hIL-6 affected regenerative pathways differently. While PTEN$^{-/-}$ expectedly activated PI3K/AKT/mTOR only in transduced neurons, AAV2-hIL-6 induced JAK/STAT3-signaling additionally in adjacent cortical neurons (Fig. 1) and raphe neurons via the transneuronal route. Therefore, it is conceivable that the combinatorial PTEN$^{-/-}$ and hIL-6 treatment enabeled stronger CST-regeneration than each treatment by itself as previously shown in the optic nerve[16–18]. However, these synergistic effects were limited to CST-regeneration due to the restricted effect of PTEN$^{-/-}$ on virally transduced cortical motoneurons. Moreover, the finding that hIL-6 alone did not measurably affect PI3K/AKT/mTOR activity but showed even stronger effects than PTEN$^{-/-}$ suggests that extensive activation of AKT/mTOR is not essential to reduce axonal dieback or improve CST regeneration. AAV2-hIL-6 did not activate the MAPK/ERK pathway either, leading to the conclusion that its beneficial effect on axon regeneration was mediated by STAT3 activation, as previously shown in the visual system[12,16,17,47]. Future experiments need to address whether a knockout or inhibition of STAT3 in cortical neurons compromises the hIL-6 effects on CST-regeneration. It is also possible that the

effects of hIL-6 on axon regeneration can be further increased in combination with a SOCS3 knockout, which usually limits the activity of JAK/STAT3-signaling[16,17].

Cortical AAV2-hIL-6 injection induced STAT3 activation and promoted axon regeneration in the CST and RpST. Although AAV2-hIL-6 enabled longer CST-regeneration than PTEN$^{-/-}$, our data suggest that the functional recovery mostly depended on the improved regeneration of serotonergic fibers. This is because (i) the PTEN$^{-/-}$ induced CST regeneration did not enable hindlimb recovery, (ii) AAV2-hIL-6 improved RpST-regeneration and functional recovery similarly in BL6 and Ola-PTEN-floxed mice; however, the effect on CST regeneration was different in these mouse strains and (iii) selective elimination of serotonergic fibers by DHT[24,27,28] abolished most of the recovered locomotion after AAV2-hIL-6 treatment. Moreover, RpST regeneration is essentially involved in locomotor recovery after less severe spinal cord injuries, which allow some endogenous sprouting of serotonergic axons[24,48–51]. Although our data demonstrate that AAV2-hIL-6-mediated functional recovery depends on the regeneration of serotonergic fibers, we cannot exclude the possibility that hIL-6 might have also stimulated axon regeneration in other neurons that receive collateral input from cortical motoneurons. Neurons in the red nuclei are such examples[35], where we also observed STAT3 activation

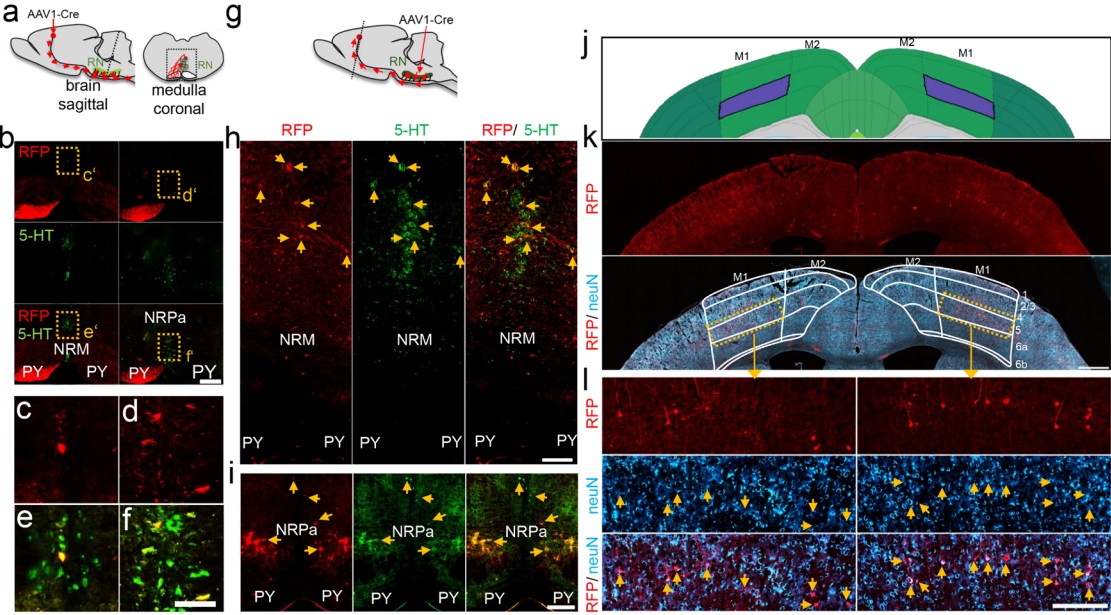

**Fig. 7 CST axon collaterals project to raphe nuclei. a** Schematic illustration, indicating the cortical application, anterograde axonal transport, and terminal release of AAV1-Cre into raphe nuclei (RN, green) in Rosa-tdTomato reporter mice (sagittal view). A coronal view of the medulla illustrates RFP-positive pyramidal (PY) axon sprouts from AAV1-Cre transduced cortical neurons. Some pyramidal axons project to raphe neurons (green), leading to transneuronal transduction and subsequent RFP expression. The dotted box indicates the area of tissue sections shown in (**b**). **b** Coronal medullary sections from Rosa-tdTomato mice 2 weeks after intracortical AAV1-Cre injection as described in (**a**). Transneuronally transduced serotonergic neurons of the nucleus raphe magnus (NRM, left) and nucleus raphe pallidus (NRPa, right) were identified by 5-HT staining (green) and RFP fluorescence (red). Scale bar: 250 μm. **c–f** Higher magnification of dashed yellow boxes as indicated in **b**. Scale bar: 100 μm. **g** Schematic illustration showing AAV1-Cre injection into raphe nuclei of Rosa-tdTomato reporter mice, and retrograde axonal transport to the motor cortex via pyramidal sprouts. **h, i** Immunohistochemical staining of brain stem sections against serotonin (5-HT, green) at the site of AAV1-Cre application as described in (**g**) 2 weeks after injection. Transduced cells in the nucleus raphe magnus (NRM, **h**), or the nucleus raphe pallidus (NRPa, I) are labeled by expression of dt tomato (RFP, red). Scale bar: 200 μm. **j** Reference map from the Allen Brain Atlas showing the primary motor cortex (M1) layer V in purple. **k** Coronal cortical section showing RFP (red) fluorescence and neuN (blue) immunostaining. The image is superimposed by the cortical map, as shown in (**j**), indicating the primary (M1) and secondary (M2) motor area and cortical layers 1–6. Scale bar: 500 μm. **l** Higher magnification of dashed boxes as indicated in (**k**), showing retrogradely transduced cortical layer 5 motor neurons expressing RFP. Scale bar: 250 μm. Representative immunohistochemical stainings shown in (**b, h–i, k–l**) were repeated three times with individual biological samples with similar results. Source data are provided as a Source data file.

following intracortical AAV2 hIL-6 application. Whether the regeneration of fibers in the rubrospinal tracts is also contributing to the beneficial effects of AAV-hIL-6 is currently unknown. Similarly, it is unclear whether cortical AAV2-hIL6 application transneuronally stimulated regeneration of long and/or short descending propriospinal interneurons, which could facilitate the bridging of signals to second-order motoneurons. However, a main role of these neurons in this context appears unlikely, as even robust propriospinal axon regeneration does not result in functional recovery after spinal cord crush[52].

Despite PTEN$^{-/-}$ induced CST regeneration alone being insufficient, improved regeneration of the CST in addition to the RpST could have contributed to the functional recovery effect of hIL-6. This would explain the finding that the more robust CST regeneration in the AAV2-hIL-6/PTEN$^{-/-}$ group compared to hIL-6 animals correlated with a slightly higher BMS although the RpST regeneration remained similar. So improved CST-regeneration might only affect subtle aspects of functional recovery on top of basic, RpST dependent walking behavior. Accordingly, CST axon regeneration reportedly improves voluntary movements and skilled locomotion in less severe pyramidotomy- and contusion-injury models, which leave other spinal tracts intact[26,40,53,54]. Future experiments need to investigate to what extent regeneration of serotonergic fibers alone can mimic the full AAV2-hIL-6 effect on functional recovery and to what extent hIL-6 improves regeneration/recovery in less severe

spinal cord injury models (e.g., pyramidotomy, hemisection, or contusion).

Although the collateral projection of cortical motoneurons into various brain areas, e.g., in the striatum, the thalamus[52,55,56], the red nucleus, and the reticular formation, are well documented[57–59], the innervation of raphe nuclei has not yet been directly shown. The current study used AAV1, which contrary to AAV2 can trans-synaptically transduce neurons[36]. Cortical AAV1 application in transgenic reporter mice demonstrated that cortical neurons project almost equally into the ipsi- and contralateral sides of the raphe nuclei. Moreover, we observed transneuronal transduction over the whole rostrocaudal length of the raphe nuclei, even though the overall rate of transneuronal transduction was expectedly low. This is due to the fact that AAVs cannot reproduce themselves without specific helper plasmid in neurons. Therefore, only virus particles that remain functionally intact after entering the cell, being directly transported along the axon and released at the synapses can transduce other neurons. Thus transneural transduction is limited. Finally, the activation of STAT3 (pSTAT3) in serotonergic raphe neurons by unilateral AAV2-hIL-6 application in the motor cortex provides further evidence for this innervation target.

The current study demonstrates that virally expressed hIL-6 is not only released from the soma to stimulate adjacent moto-neurons in a paracrine fashion but that it is also transported over long distances in axons of either RGCs in the visual system or

cortical neurons. In contrast to AAV1 mediated transneuronal transduction, hIL-6 is continuously expressed, transported, and released at axon terminals to secondary neurons as reflected by a high percentage of pSTAT3 positive serotonergic neurons (~50%) in all raphe nuclei. The release of active hIL-6 at axonal terminals/ synapses was verified in cell culture experiments and by pSTAT3 staining in visual target areas and directly in the raphe nuclei themselves. In this context, it is worth mentioning that hIL-6 is highly potent, so that even the smallest quantities can activate different types of neurons expressing gp130, the receptor to which it directly binds[18]. Moreover, due to the localization of the raphe nuclei in the medial column and, therefore, in the center of the brainstem, one motor cortex hemisphere projects into both raphe nuclear hemispheres along the whole rostrocaudal length. Together with the high potency of hIL-6, this explains why the unilateral cortical application of AAV2-hIL-6 improved both hindlimbs' recovery to a similar degree and why the bilateral application had no additional beneficial effect on RpST regeneration or functional recovery.

In conclusion, our finding that the transneuronal application of hIL-6 enables functional recovery opens possibilities to further improve the functional outcome by combining it with other strategies, such as neutralizing extracellular inhibitors at the lesion site[8,50,60] or bridging the lesion site with permissive grafts[61,62]. Such combinatorial approaches, also in less severe injury models, may maximize axon regeneration and functional recovery after spinal cord injury, potentially also in humans.

## Methods

**Mouse strains**. Male and female mice were used for all experiments. Wild-type C57BL/6 and 129/Ola mice were crossbred with PTEN[f/f] mice (C57BL/6;129) to obtain C57BL/6;129/J-TgH (Pten-flox) animals without further crossbreeding. C57BL/6J mice were obtained from Janvier Labs. Rosa26 loxP-stop-loxP-tdTomato (Rosa-tdTomato) mice were obtained from Jackson Laboratories (Stock No: 007914). All animals were housed under the same conditions for at least ten days before the start of experiments and generally maintained at 24 °C ambient temperature and 60% humidity with a 12 h light/dark cycle and ad libitum access to food and water. All experimental procedures were approved by the local animal care committee (LANUV Recklinghausen) and conducted in compliance with federal and state guidelines for animal experiments in Germany.

**Intracortical injection of pups**. Postnatal (P1) PTEN floxed (C57BL/6;129/J-TgH (Pten-flox)) pups were fixed on a stereotactic frame and continuously supplied with 2% isoflurane for anesthesia via a mouthpiece. A midline incision into the skin was made to expose the skull using microscissors. Since the skulls of P1 mice are still soft, the cortex could be accessed using a 30-gauge needle to create two small holes in the left skull hemisphere with the following coordinates: −0.2 mm and 0.3 mm anteroposterior, 1.0 mm lateral to bregma. For AAV2-GFP or AAV2-Cre application, 770 nl of the virus suspensions were injected at a depth of 0.5 mm into the two sites using a pulled glass pipette connected to a nanoliter injector (Nanoject II, Drummond). To inject 770 nl, we applied 11 pulses of 70 nl at a rate of 23 nl/s and waited for 10 s after each pulse to allow distribution of the virus solution. After injection, the pipette was left in place for 1 min before being carefully withdrawn, and the incision carefully closed with a 4-0 black silk suture.

**Intracortical injection of adult mice**. For intracortical injections, adult PTEN floxed (C57BL/6;129/J-TgH(Pten-flox)) or adult wt (C57BL/6) mice were anesthetized by intraperitoneal injection of ketamine (120 mg/kg) and xylazine (16 mg/kg) and then placed in a stereotaxic frame. A midline incision was made over the skull to open the skin and to reveal bregma. A micro drill with a 0.5 mm bit was used to open a 2 × 2 mm window on each side of the skull to expose the sensorimotor cortex. The respective AAV2 was injected 30 min after spinal cord injury into the cortex layer V through a glass pipette attached to a nanoliter injector (Nanoject II, Drummond). To this end, four injections of 490 nl each were given either unilaterally (left hemisphere) or in both hemispheres, at the following coordinates: 1.5 mm lateral, 0.6 mm deep, and 0.5 mm anterior; 0.0 mm, 0.5 mm, and 1.0 mm caudal to bregma. For injecting 490 nl into each injection site, we applied 7 pulses of 70 nl at a rate of 23 nl/s and waited for 10 s after each pulse to allow distribution of the virus solution. The needle was left in place for 1 min before moving to the next site, and the brain was kept moist during the procedure by moving the skin over the exposed area after each injection. After surgery, the skin was closed with sutures. The virus transduced mainly neurons in layer 5 of the primary motor cortex (M1). As determined by GFP coexpression, the AAV-hIL-6,

for example, transduced on average 1630 (±124 SEM) cortical cells. We observed only very rare transduction of astrocytes or neurons of other M1 layers (Fig. S1a–e). All GFP- expressing neurons also expressed hIL-6 (Fig. S1f). Therefore, hIL-6 transduction was identified by GFP expression during the whole study.

**Injection into the red nucleus**. Adult wt mice were anesthetized by intraperitoneal ketamine (100 mg/kg) and xylazine (10 mg/kg) injection and then placed in a stereotaxic frame. A midline incision was made over the skull to reveal bregma. A micro drill with a 0.5 mm bit was used to open a 1 ×1 mm window in the skull to expose the cortex. Biotinylated dextran amine (BDA, 10,000 MW, 10% solution in water, Invitrogen, D1956) was injected into the red nucleus through a glass pipette attached to a nanoliter injector (Nanoject III, Drummond). To this end, 500 nl was given at the following coordinates[26]: 0.6 mm lateral, 3.5 mm deep, and 2.5 mm caudal to bregma. For injecting 500 nl, we applied 5 pulses of 100 nl at a rate of 5 nl/s and waited for 10 s after each pulse to allow for distribution of the virus solution. The needle was left in place for 1 min before removal, and the brain was kept moist during the procedure. After surgery, the skin was closed with sutures.

**Spinal cord crush**. For complete spinal cord crush, adult PTEN floxed (C57BL/6;129/J-TgH(Pten-flox)) or wt (C57BL/6) mice were anesthetized by intraperitoneal injections of ketamine (120 mg/kg) and xylazine (16 mg/kg). A midline incision of ~1.5 cm was performed over the thoracic vertebrae. The fat and muscle tissue were cleared from thoracic vertebrae 7 and 8 (T7, T8). While holding onto T7 with forceps, we performed a laminectomy at T8 to expose the spinal cord. Afterward, the complete spinal cord was crushed for 2 s with forceps that had been filed to a width of 0.1 mm for the last 5 mm of the tips to generate a homogeneously thin lesion site. To ensure that the spinal cord's full width was included, we took care to gently scrape the forceps' tips across the bone on the ventral side of the vertebral canal so as not to spare any axons ventrally or laterally. After surgery, the muscle layers were sutured with 6.0 resorbable sutures, and the skin was secured with wound clips. The completeness of the injury in the SCC model was verified by an astrocyte free gap and the absence of any spared CST or raphe spinal tract (RpST) fibers caudal to the lesion site shortly after injury (Fig. S13e–j). Two animals with PTEN[−/−] and one control mouse which had received AAV-GFP were excluded because they did not meet the criteria described above.

**CST tracing**. To trace the axons of cortical motoneurons in PTEN floxed (C57BL/6;129/J-TgH(Pten-flox)) or wt (C57BL/6) mice, we injected the axon tracer biotinylated dextran amine (BDA, 10,000 MW, 10% solution in water, Invitrogen, D1956) into the sensorimotor cortex 2 weeks before the mice were sacrificed[10,11,41]. Therefore, the skin was opened, and 490 nl BDA was applied to each injection site using the same procedure and coordinates as described for the AAV2 injections. After surgery, the skin was closed with sutures. In groups with the PTEN knockout, we verified that 70-80% of BDA-traced neurons were also Cre-positive (Fig. S13a–d).

**DHT injection**. Mice received bilateral intracerebroventricular injections of 30 μg of the serotonin neurotoxin 5,7-dihydroxytryptamine (DHT, Biomol) dissolved in 0.5 μl of 0.2% ascorbic acid in saline to deplete serotonergic inputs to the lumbar spinal cord. To this end, mice were anesthetized and fixed under a stereotactic frame as described above for intracortical injections. The tip of a glass micropipette attached to a nanoliter injector (nanoject II, Drummond) was positioned at the following coordinates: 0.6 mm posterior, 1.6 mm lateral to bregma, and 2 mm deep from the cortical surface. DHT was injected at the same rate as described for the AAV2 injections, and the pipette was left in place for 1 min before withdrawal. The monoamine uptake inhibitor, desipramine (Sigma), was administered at 25 mg/kg intraperitoneally 30 min in advance for all mice that received DHT to prevent the toxin's uptake into noradrenergic neurons.

**Intraocular injection of AAV2**. Adult mice (C57BL/6) received intravitreal AAV2 injections (1 μl)[18,63]. After 3 weeks, animals were sacrificed and tissues isolated.

**Injection into the raphe nuclei**. For brainstem injections, adult Rosa26 loxP-stop-loxP-tdTomato mice were anesthetized by intraperitoneal ketamine (120 mg/kg) and xylazine (16 mg/kg) injections and then placed in a stereotactic frame. A midline incision was made over the skull to reveal bregma. A micro drill with a 0.5 mm bit was used to open a 2 × 2 mm window in the center of the skull ~5 mm caudal to bregma to expose the cerebellum. AAV1-Cre obtained from Addgene was injected through a glass pipette attached to a nanoliter injector (Nanoject III, Drummond). To this end, 300 nl injections were given to five sites spanning the whole rostrocaudal length of the brainstem raphe nuclei. The following coordinates were used: 0 mm lateral, 5.2 mm deep, and 4.7 mm, 5 mm, 5.3 mm, 5.6 mm, and 5.9 mm caudal to bregma. For injecting 300 nl into each injection site, we applied six pulses of 50 nl at a rate of 10 nl/s and waited for 10 s after each pulse to allow distribution of the virus solution. The needle was left in place for 1 min before moving to the next site, and the brain was kept moist during the procedure by moving the skin over the exposed area after each injection. After surgery, the skin was closed with sutures.

**AAV virus production**. AAV1-Cre virus was obtained from Addgene (pAAV.CMV.HI.eGFP-Cre.WPRE.SV40; plasmid #105545), having a titer of $2 \times 10^{12}$ GC/ml. AAV2 viruses were produced in our laboratory using a pAAV-IRES-eGFP expression vector for hIL-6[18]. A plasmid for Cre expression was kindly provided by Dr. Zhigang He (Children's Hospital, Boston)[64]. AAV plasmids carrying either cDNA for Cre-HA, GFP, or hIL-6 downstream of a CMV promoter were co-transfected with pAAV-RC (Stratagene) encoding the AAV genes rep and cap, and the helper plasmid (Stratagene) encoding E24, E4, and VA into AAV-293 cells (Stratagene) for recombinant AAV2 generation. Purification of virus particles was performed as described previously[12]. The titer of the AAVs ranged from $1.5 \times 10^{12}$– $3 \times 10^{12}$ GC/ml. Mainly cortical neurons are transduced upon intracortical injection of AAV2 into layer V, as this virus serotype is highly neurotropic.

**Tissue processing**. Mice were anesthetized and perfused through the heart with cold PBS followed by 4% PFA in PBS. Isolated brains and spinal cords were postfixed in 4% PFA overnight, then transferred into 30% sucrose at 4 °C for 5 days. The bulk of the brain, medulla, and different spinal cord segments was embedded in Tissue-Tek (Sakura) and frozen at −20 °C. For spinal cords subjected to T8 crush, a ~1.2 cm segment from 3 mm rostral to, and 9 mm caudal from the injury was embedded for sagittal sections. Spinal cord tissues ~2 mm in length just rostral to and caudal from this 1.2 cm segment were embedded for transverse sections to evaluate axon sprouting proximal to the lesion site (rostral segments) and completeness of the lesion by identification of potentially spared axons (caudal segments). Tissues were sectioned on a cryostat (Leica) at a thickness of 20 µm, thaw-mounted onto Superfrost plus slides (Thermo Fisher) and stored at −20 °C until further use. Transverse sections of the medullary pyramids ~1 mm above the pyramidal decussation were cut at a thickness of 14 µm to quantify the total number of BDA traced corticospinal axons (see below).

**DRG neuron two-compartment cultures**. Dorsal root ganglion-(DRG) neurons were harvested from adult mice[65]. Isolated DRG (T8-L5) were incubated in 0.25% trypsin/EDTA (GE Healthcare) and 0.3% collagenase Type IA (Sigma) dissolved in DMEM (Invitrogen) at 37 °C and 5% $CO_2$ for 45 min followed by mechanical dissociation. Cells were resuspended in DMEM containing B27-supplement (Invitrogen; 1:50) and penicillin/streptomycin (500 U/ml; Merck, Millipore) and seeded into the somal compartment of microfluidic two-compartment chambers (AXIS Axon Isolation Device, Millipore), mounted on poly-D-lysine (0.1 mg/ml, molecular weight 70,000-150,000 kDa; Sigma) plus laminin-coated (20 µg/ml; Sigma) culture dishes according to the manufacturer's instructions. Neurons were cultured at 37 °C and 5% $CO_2$ for 3 days to allow axon extension through the microchannels and then transduced with hIL-6 expressing baculoviruses (BV) or GFP expressing control BV. Diffusion of particles or proteins through the microchannels into the axonal compartment was prevented by hydrostatic pressure established by using two different volumes of medium with a resulting antagonistic microflow of liquid between the axonal and the somal chambers. After 24 h, the virus-containing medium was replaced with fresh culture medium (DMEM, 1:50 B27 supplement, and 1:50 penicillin/streptomycin). After 48 h, the medium of 5 axonal compartments from either BV-GFP or BV-hIL-6 transduced cells were collected respectively and concentrated 10-fold using Merck Amicon centrifugal filter units with a retaining size of 30 kDa. The concentrated medium was then used to culture retinal ganglion cells (RGCs), as described in the following paragraph. After fixation with 4% PFA for 30 min at RT, cell cultures were processed for immunocytochemical staining with antibodies against synapsin (1:1000, Millipore, RRID:AB_2200400), GFP (1:500; Novus; RRID:AB_10128178), and IL-6 (1:500; Abcam; RRID:AB_2127460). Stained axons were photographed using a confocal microscope (630×, SP8, Leica).

**Dissociated retinal cell cultures**. For low-density retinal cell culture preparation, tissue culture plates (4-well plates; Nunc, Wiesbaden, Germany) were coated with poly-D-lysine (PDL, 0.1 µg/ml, molecular weight between 70,000 and 150,000 Da; Sigma, St Louis, USA) and with 20 mg/ml laminin (Sigma). Mice were killed by cervical dislocation. Retinae were rapidly dissected from the eyecups and incubated at 37 °C for 30 min in a digestion solution containing papain (10 U/mL; Worthington) and L-cysteine (0.2 µg/mL; Sigma) in Dulbecco's modified Eagle medium (DMEM; Thermo Fisher). Retinae were then rinsed with DMEM and triturated in 1.5 ml of medium obtained from axonal compartments of DRG neuron cultures, as described above. Dissociated cells were passed through a cell strainer (40 µm, BD Falcon, Franklin Lakes, USA), and 300 µl of cell suspension was added into each well of the coated culture plates. Retinal cells were cultured for 72 h and then fixed with 4% PFA (Sigma).

After fixation with 4% PFA for 30 min at RT, cell cultures were processed for immunocytochemical staining with a βIII-tubulin antibody (1:2,000, BioLegend, RRID:AB_2313773). All RGCs with regenerated neurites were photographed using a fluorescence microscope (×200, Axio Observer.D1, Zeiss), and neurite lengths were determined using ImageJ software. The total number of βIII-tubulin-positive RGCs with an intact nucleus (6-diamidino-2-phenylindole (DAPI)) per well was quantified to test for potential neurotoxic effects. Cultures were arranged in a pseudo-randomized manner on the plates so that the investigator would not be aware of their identity. The average neurite length per RGC was determined by

dividing the sum of neurite lengths per well by the total number of RGCs per well (on average ~500 RGCs). The resulting neurite length per RGC also includes most RGCs, which did not produce any neurites (~90%), thereby representing neurite growth and neuritogenesis. Thus, the obtained values are much lower than the average axon length of RGCs with extended neurites (~20–500 µm). Data were presented as normalized means ± SEM of six independent experiments, each with four wells per treatment as technical replicates.

**Immunohistochemistry**. Cryosections of brains, medullas and spinal cords were thawed, washed in PBS for 15 min and permeabilized by 10 min incubation in 100% Methanol (Sigma). Sections were stained with antibodies against the HA-tag (1:500; Sigma; RRID:AB_260070), pSTAT3 (1:200; RRID: AB_2491009), pAKT-Thr308 (1:500; RRID: AB_2629447), pERK1/2 (1:500; RRID: AB_331646), pS6 (1:500; RRID: AB_2181035) (all Cell Signaling Technology), GFP (1:500 Thermo Fischer, RRID: AB_221570), GFP (1:500; Novus; RRID:AB_10128178), IL-6 (1:500; Abcam; RRID:AB_2127460) GFAP (1:50; Santa Cruz Biotechnology; RRID: AB_627673), serotonin (1:5,000; Immunostar; RRID: AB_572263, RRID: AB_572262) and NeuN (1:2,000; Abcam, AB_10711040). Secondary antibodies included anti-mouse, anti-goat and anti-rabbit conjugated to Alexa Fluor 405 (1:500; Jackson ImmunoResearch), 488, or 594 (1:1,000; Invitrogen), respectively. BDA traced CST neurons and axons were detected using streptavidin fluorescently labeled with Alexa Flour 405, 488, or 594 (1:500, Thermo Fischer). Sections were embedded in Mowiol (Sigma) and analyzed using fluorescence widefield (Observer.D1, Zeiss; Axio Scan.Z1, Zeiss, VS120, Olympus), or confocal laser scanning (SP8, Leica) microscopes. Images were processed using the following software: Axiovision 4.9 Zeiss; LASX 3.5.1 Leica; Virtual Slide 2.9 Olympus; ZEN black 2.3 Zeiss; ImageJ 1.51f (https://imagej.nih.gov/ij/); Adobe Photoshop CS5 version 12.

**Western blot analysis**. For cortical lysate preparation, mice were killed, and a cuboid piece of cortical tissue of $1.5 \times 2$ mm with 1 mm depth from the cortical surface was dissected and isolated from the sensorimotor cortex around the coordinates used for viral injection. For lysates of brain stem raphe nuclei, a 2.5 mm long and 1.5 mm wide piece of tissue was isolated around the midline of the medulla with a depth of 1 mm starting above pyramidal tracts, which were thereby also removed. Tissues were homogenized in lysis buffer (20 mM Tris-HCl, pH 7.5, 10 mM KCl, 250 mM sucrose, 10 mM NaF, 1 mM DTT, 0.1 mM $Na_3VO_4$, 1% Triton X-100, 0.1% SDS) with protease inhibitors (Calbiochem) and phosphatase inhibitors (Roche) using five sonication pulses at 40% power (Bandelin Sonoplus). Lysates were cleared by centrifugation in a tabletop centrifuge (Eppendorf) at $4150 \times g$ for 10 min at 4 °C. Proteins were separated by SDS/PAGE, using Mini TGX gels (10%, Bio-Rad) according to standard protocols, and then transferred to nitrocellulose membranes (0.2 µm; Bio-Rad). Blots were blocked in 5% dried milk in Tris- or phosphate-buffered saline solution with 0.05% Tween-20 (TBS-T and PBS-T, respectively; Sigma) and incubated with antibodies against total STAT3 (1:1000; RRID:AB_2629499), pSTAT3 (1:1000; RRID: AB_2491009), pAKT-Thr308 (1:500; RRID: AB_2629447), pERK1/2 (1:500; RRID: AB_331646), pS6 (1:500; RRID: AB_2181035) (all Cell Signaling Technology), GFP (1:500 Thermo Fischer, RRID: AB_221570), IL-6 (1:1000; Abcam; RRID:AB_2127460), or against β-actin (1:5000; AC-15; Sigma, RRID: AB_476744) at 4 °C overnight. All primary antibodies were diluted in TBS-T or PBS-T containing 5% BSA (Sigma). Anti-rabbit or anti-mouse IgG conjugated to HRP (1:80,000; Sigma) were used as secondary antibodies. Primary antibodies against pSTAT3 and pAKT were detected using a conformation-specific HRP-conjugated anti-rabbit IgG secondary antibody (1:4,000; Cell Signaling Technologies; RRID: AB_10892860). Antigen-antibody complexes were visualized using an enhanced chemiluminescence substrate (Bio-Rad) on a FluorChem E detection system (ProteinSimple). Western blots were repeated using lysates from different animals as independent biological replicates. Each blot contained samples of a control group, which was used as a reference for data normalization. For this purpose, control samples were either loaded individually or mixed in equal proportions. Pixel intensities of bands were densitometrically quantified using ImageJ software and normalized to the respective loading control intensities. Moreover, in each replicate, treatment groups were normalized to the relative intensity of the control group. Individual blots used for quantification are shown in Figs. S14, S15.

**Tissue clearing**. Brain and spinal cord tissue were isolated from mice after perfusion with PBS, and 4% PFA as described above. Tissue was postfixed overnight at 4 °C before further processing. For whole-mount immunostaining, tissue was washed two times for 1 h in PBS containing 0.2% Triton-X-100 and then incubated overnight at 37 °C in PBS containing 0.2% TritonX-100 and 20% DMSO. Afterward, the tissue was again incubated overnight at 37 °C in PBS containing 0.1% Tween-20, 0.1% Triton-X-100, 0.1% deoxycholate, 0.1% NP40 and 20% DMSO. After two washing steps in PBS with 0.2% TritonX-100, specimens were permeabilized (PBS + 0.2% TritonX-100 + 23% glycine + 20% DMSO) for 1 day at 37 °C, blocked (PBS + 0.2% TritonX-100 + 6% donkey serum + 10% DMSO) at 37 °C for 1 d, and incubated with primary antibodies against pSTAT3 (1:100; Cell Signaling Technology; RRID: AB_2491009, RRID:AB_2198588), GFP (1:500; Thermo Fischer, RRID: AB_221570), serotonin (1:2500; Immunostar; RRID: AB_572263, RRID: AB_572262) and NeuN (1:500; Abcam, RRID: AB_10711040) at 37 °C in

blocking solution for 2 d. Afterward, tissue was incubated for another 2 d at 37 °C with fluorescently labeled streptavidin (Alexa Flour 405, 488, or 594; 1:250; Thermo Fischer) to detect BDA traced CST axons, or with secondary antibodies including anti-mouse, anti-goat, and anti-rabbit conjugated to Alexa Fluor 405 (1:250; Jackson ImmunoResearch), 488 or 594 (1:500; Invitrogen), respectively. Subsequently, spinal cords and brains were dehydrated in ascending concentrations of tetrahydrofuran (50%, 80%, 100%), incubated in dichloromethane and then transferred to a clearing solution (benzyl alcohol and benzyl benzoate (1:2)) and imaged with a confocal laser scanning microscope (SP8, Leica).

**Quantification of the total number of BDA traced CST axons.** To obtain the number of BDA traced axons, four adjacent medullary cross-sections, ~1 mm rostral to the pyramidal decussation, were stained with fluorescently labeled streptavidin and imaged with a fluorescence microscope. Four $25 \times 25$ μm squares were randomly superimposed on each image, and the number of labeled axons in every square was counted and averaged. The entire pyramid area and the total area inside a counted square were measured and used to extrapolate the total number of BDA-labeled axons per pyramid.

**CST sprouting proximal to the lesion site.** The sprouting index was determined as described previously[10,53]. To this end, images of transverse spinal cord sections ~3 mm rostral to the lesion site from untreated controls and mice 8 weeks after spinal cord crush were taken. To quantify the number of sprouting axons, we drew a horizontal line through the central canal and across the gray matter's lateral rim. Three vertical lines were then drawn to divide the horizontal line into three equal parts (Z1, Z2, Z3), starting from the central canal to the lateral rim. Only fibers crossing these lines were counted in each section. The results were presented after being normalized against the number of total CST fibers at the medulla level obtained as described above. At least three sections were quantified for each mouse.

**Evaluation of CST-axon retraction.** The retraction of axons rostral to the injury site was quantified as an axon density index by measuring the BDA staining intensity as described previously[10,11,41]. For BDA staining quantification, series of 100 μm wide and 2 mm long rectangles covering the entire dorsoventral axis of the spinal cord were superimposed onto sagittal sections, starting from 1.5 mm rostral up until the injury site, which was visible by its morphology and additional GFAP staining of the glial scar. The mean gray value of the background signal in BDA negative areas was measured in each spinal cord section and subtracted from later quantified BDA staining. To quantify BDA staining, the mean gray values within superimposed rectangles were measured in 100 μm steps from 1.5 mm to 0 mm rostral to the lesion site. After subtracting the background, values were divided by the mean gray value within the rectangle at 1.5 mm rostral to the lesion site. At this distance, BDA staining intensity represents the total amount of labeled axons in the main CST since retraction is usually limited to a distance of ~0.5–1 mm proximal to the lesion site. By quantifying and averaging all consecutive sagittal sections through the spinal cord containing the main CST from each animal, we received the axon Density Index, which was then plotted as a function of the distance to the injury ($-1.5$ mm–0 mm).

**Evaluation of CST regeneration.** Sagittal sections of spinal cord segments starting from 3 mm caudal to and up until 9 mm distal to the lesion site were stained for BDA using fluorescently labeled streptavidin as described above. Although the injury site was visible by morphology, at least one section per animal was stained against GFAP to ensure the lesion edges' identity. Axon regeneration was quantified from stitched images scanned under a ×10 objective on a fluorescence Axio Scan.Z1 microscope (Zeiss). A grid with lines spaced every 100 μm was aligned to each image, and the number of axons at 1, 5, 6, and 7 mm from the caudal edge of the lesion were quantified in ~40 adjacent sections per animal. For quantification, all axons observed within an area of 0.5 mm caudal and rostral to the respective measuring point were counted. To obtain the axon index, the total number of BDA labeled regenerated CST axons was quantified at each distance over all consecutive sections of each spinal cord and then divided by the total number of BDA labeled axons in the medulla as described above and displayed in Fig. S2b.

**Evaluation of RpST fiber regeneration.** As described above, sagittal spinal cord sections were stained against serotonin and imaged under a fluorescence microscope (×10, Axio Scan.Z1, Zeiss). Images were superimposed with a grid as described for CST regeneration. Axons that reached the indicated distances from the lesion site's distal border were quantified in ~40 sections (every second section) per animal, averaged over all analyzed sections, and then extrapolated to the whole number of sections per spinal cord. For quantification, all axons observed within an area of 0.5 mm caudal and rostral to the respective measuring point were counted.

**Evaluation of macrophage/microglia activation.** For quantification of macrophage/microglial activation in the thoracic spinal cord 8 weeks after intracortical AAV2-gfp or AAV2-hIL-6 injection and T8 spinal cord crush, cross-sections from mice were immunohistochemically stained against CD11b (1:500, BioRad, RRID: AB_324660) and then imaged (Olympus VS120). For each section, the mean grey

value of the CD11B staining was obtained using ImageJ software. In this way, 3 sections per animal were analyzed, values then averaged per animal, and then among the respective treatment group. Afterward, the measured signal was normalized to the untreated control group's staining intensity to obtain the "CD11b staining index".

**Quantification of DHT-mediated loss of raphespinal neurons.** For quantification of DHT-induced loss of serotonergic neurons, the coronal medullary sections with the nucleus raphe magnus, nucleus raphe pallidus, and nucleus raphe obscurus were analyzed from untreated control animals and mice which had received intracerebroventricular DHT treatment 1 day before, as described above. Sections were then immunohistochemically stained with a serotonin (5-HT) antibody and imaged using a fluorescence microscope (Olympus VS120). Raphe nuclei were located according to the Allen Brain Atlas using the pyramids and the pyramidal decussation as landmarks. Both structures were visible under brightfield illumination. The number of 5-HT positive cells per section was counted in four sections per animal taken from different regions over the entire rostrocaudal length of the respective raphe nuclei. Only neurons with an intact nucleus (DAPI staining) were used for the quantification. These values were averaged for each animal and then averaged among the single raphe nuclei's respective treatment group.

To analyze the loss of raphespinal axons, thoracic spinal cord cross-sections from DHT-treated and untreated animals were stained against serotonin and then imaged (Olympus VS120). For each section, the mean grey value of the staining was obtained using ImageJ software. The mean grey value of the background signal measured in respective unstained cross-sections was subtracted to reduce bias. Afterward, the measured signal was normalized to the untreated control group's staining intensity to obtain the "5-HT staining index". In this way, four sections per animal were analyzed, then averaged per animal, followed by averaging among the respective treatment group.

**Basso mouse scale (BMS).** The locomotory behavior of mice was tested and scored according to guidelines of the Basso Mouse Scale[21]. Therefore, each mouse was placed separately in a round open field of 1 m in diameter and observed by two testers for 4 min. The open-field arena was surrounded by a 30 cm high transparent acrylic glass plane, which prevented the mice from escaping but still allowed observation from the side view. During testing, the investigators were blinded to the identity of the cohorts. Animals from different treatment groups were randomly distributed between cages. Scoring was based on different parameters such as ankle movements, paw placement, stepping pattern, coordination, trunk instability, and tail position, with a minimum score of 0 (no movement) to a maximum score of 9 (normal locomotion). For animals that have attained frequent plantar stepping (BMS ≥ 5), we also determined the BMS subscore, which discriminates more precisely between the fine details of locomotion, which may not be differentiated by the standard BMS score[11,21]. The BMS subscore applies only for animals that achieved at least frequent stepping (BMS score ≥5). It is designed to detect differences between groups that plateau at a similar level on the main scale. The calculation is based on the same score sheets, but it emphasizes refined distinctions in stepping. To this end, stepping-related features receive numeric values in advanced categories like the consistency of plantar stepping, degree of coordination between hind- and forelimbs, paw position (parallel or rotated), degree of trunk instability, and tail position (up or down). In this way, a cumulative score can range from a minimum of 0 up to a maximum of 11.

Before testing, mice were acclimated to being handled and to the open field environment. BMS tests were performed before the injury, on days 1, 3, 7, and then weekly (over eight weeks) after spinal cord injury. A BMS score of 0 at 1 d after injury and the absence of BDA staining in distal spinal cord cross-sections, as described above, were used as quality criteria for completeness of the lesion. Mice that did not meet these criteria were excluded from the analysis.

**Catwalk gait analysis.** The CatWalk XT system and respective Catwalk XT 10.6 software (Noldus) were used for automated evaluation of footprints, basal motor behavior, and coordination during continuous locomotion along a defined walkway. The Catwalk system is commonly used to analyze functional recovery after spinal cord injury[66–71] and consists of a narrow corridor with a glass plate illuminated by a green LED light and a high-speed camera positioned below. An additional illuminated ceiling with a red LED light makes the body contour of the animal visible. Paws touching the glass plate are illuminated by green light at the contact area only, producing a footprint detected by the camera. Afterward, Cat-Walk XT software identifies changes in the dimensions, position, and timing of each footprint. Catwalk testing was performed after the final BMS evaluation (8 weeks postinjury) of respective mice. Each animal was allowed to cross the runway and was recorded in three continuous runs from below the glass by the digital video camera. Using this system, we analyzed the following parameters:

Max area: Represents the maximum area of a paw that comes into contact with the glass plate during one step. Total area: Refers to the total surface area of the complete footprint. The total area is, by definition, at least as large as the max area. Stride length: The stride length is the distance between successive placements of the same paw. Base of support: The base of support (BOS) is the average width between the footprints of the hind paws. Regularity index: The regularity index (RI), as

determined by the CatWalk, is a measure of coordination between hind and forelimb movement. The RI is defined as RI = ((NSSP × 4)/PP) × 100% (NSSP: number of standard step sequence patterns, PP: total number of paw placements). Traversing the walkway with a RI of 100% is considered as coordinated.

**Analysis and statistics**. Male and female mice were randomly distributed among the treatment groups for all experiments. Among all studies concerning anatomical axon regeneration and functional recovery, we analyzed a total of 29 AAV-2 GFP and 39 AAV-2 hIL-6 treated wild-type animals comprising male (gfp, $n = 14$; hIL-6, $n = 14$) and female (gfp, $n = 15$; hIL-6, $n = 25$) animals. No gender-specific differences were observed in any of the analyses.

For all analyses, including BMS testing, CST regeneration/sprouting/dieback, RpST regeneration, characterization of DHT treatment, and immunohistochemistry, individual mice or samples of respective animals were randomly labeled with different letters in a way that the investigator was blinded to the identity of cohorts. For all images that were taken with the widefield fluorescence microscope (Axio Observer. D1, Axio Scan Z1, Zeiss, VS120, Olympus), we used standardized camera settings and the same exposure times among all samples. For confocal imaging (LSM 510, Zeiss; SP8, Leica), standardized settings of the resolution, z-stacks, pinhole, laser intensity, detector gain/offset were used for all comparatively captured images among all samples. The significance of intergroup differences was evaluated using Student's $t$-test or analysis of variance (ANOVA) followed by Holm-Sidak, or Tukey post hoc test using the Sigma STAT 3.1 software (Systat Software). Statistical details ($p$-values) for individual experiments are presented in the corresponding figure legends.

**Reporting summary**. Further information on research design is available in the Nature Research Reporting Summary linked to this article.

## Data availability

All data supporting the findings are included within this manuscript and its Supplementary information. Source data are provided with this paper.

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

## Acknowledgements

We would like to thank Marcel Kohlhaas, Christopher Brennsohn, and Davina Stoesser, Kessy Brzozowski, for technical support and to Dr. Daniel Terheyden-Keighley for helpful comments on the manuscript. The German Research Foundation supported this work.

## Author contributions

D.F. and M.L. designed the project. M.L., D.F., C.Z., P.G., A.A., and G.G. performed and planned the experiments. D.F., M.L., C.Z., and P.G. analyzed the data. D.F. supervised the research; D.F. and M.L. wrote the paper.

## Funding

## Competing interests

The authors declare no competing interests.
