## [Peer Review File · Nature Communications]

Reviewers' Comments:

Reviewer #1:

Remarks to the Author:

This is a comprehensive set of studies showing remarkable effects of AAV-hIL-6 injections into the sensorimotor cortex on regeneration of axons below a crush injury site. Functional recovery was shown to be dependent on regeneration of serotonergic raphespinal tracts, while release of hIL from axon connections between motor cortex and serotonergic raphe nuclei was demonstrated. Overall, this is an outstanding innovative study that reveals novel information that could potentially influence thinking in the field and contribute to the development of treatment options for spinal cord injury.

Main concerns are related to practices and reporting of rigor and allowing for complete reproducibility.

1. Many of the experiments include only low numbers of animals, with sample sizes of 3, 4, or 5. In addition, there is a lack of adequate reporting of the outcomes of the statistical analyses. Since underpowered studies are detrimental to the ability to translate or build on findings from preclinical studies, it is important that rigor of the studies is clearly demonstrated. These are the recommendations:

- Authors need to comment on the rigor of their studies in light of the low sample sizes. For example: Were power analyses conducted? Were key controls included or findings replicated from previous studies?
- Furthermore, authors need to provide all statistical details for all analyses including F values, dfs, and exact p values for all comparisons in supplementary tables. In several of the figures, variability of the data is so large that is difficult to comprehend that data passed tests for normality and weighted posthoc tests for further pairwise comparisons.
- Currently, authors differentiate between different p-values with separate symbols. However, there is no physiological relevance to differences in p values; rather lower p values usually suggest less variability in the data. It is therefore not informative nor encouraged to have different indicators for different p values in the figures and one indicator is sufficient.

2. Authors need to provide detailed information for practices that were used to eliminate bias in all analyses.

3. Authors need to expand on details in the methods section to allow for full and complete reproducibility. Below are a few examples, but authors should expand on many parts of the methods to provide all details needed for other groups to reproduce these experiments.

- Western blot analysis; how were data analyzed over the technical replicates?
- CST sprouting: what are the axon density index and axon index? Explain more clearly in methods how these were calculated and explain when first mentioned in figure legends. What exactly is displayed on the y-axis in Figure 2? it is hard to imagine that the few axons visible in Figure 2e-h divided by BDA labeled fibers in the medulla can result in the numbers shown in the graph in 2i. Please clarify.
- Evaluation of CST regeneration: it is mentioned that only animals with complete lesions were included in the evaluation. What portions of animals were excluded and what were the final sample sizes? Since the authors presumably have the behavioral data for animals without complete lesions, it may be useful to also provide the behavioral recovery of these incomplete lesions, which will add to the discussion of comparisons with other less severe injury models.
- Male and female mice were used: provide exact numbers for each study and were there sex differences in any of the analyses?
- Immunohistochemistry: what validation procedures were used for the primary antibodies and for staining protocols?
- A large amount of immunostaining was performed for labeled cells (for example pStat3), but it is unclear if any quantitative analyses were performed and thus if the images shown in the figures are representative of the data. Since western blot analyses were conducted with n=3 sample sizes,

adding quantitative analysis of numbers of immunoreactive cells would have greatly strengthened the findings. Confocal image capturing and use of standardized camera settings was not described for any analyses.

Additional comments:

Figure 7 (and in several other figures): it is very difficult to see dual labeling. Perhaps show separate images or use arrows.

Panel I, Figure 7: it appears that there are mainly BDA labeled cells and only very few axons in the red nucleus of the hIL-6-treated animal. The BDA labeling appears to be very different compared to control; is that a real observation across all animals? The images in this figure don't appear to support the finding that pSTAT3 labeled neurons are targeted by BDA fibers.

Supplemental materials: Figures and videos were very informative, except for video 7: the red signal isn't visible and it is unclear without any labeling of landmarks exactly where anatomically this video is rendered.

Reviewer #2:

Remarks to the Author:

In the manuscript by Leibinger et al, the authors use a previously described hIL6 viral application approach tested in optic nerve regeneration in spinal cord injury. They observe improved numbers of regenerating axons, particularly raphespinal axons and elevated BMS scores. They argue that a transneuronal delivery of IL6 from cortical to raphe neurons is the underlying mechanism and emphasize the advantages of their approach over regeneration effects observed in PTEN mouse mutants.

Overall, the authors show some effects on regeneration and the transneuronal mechanism might be very interesting. Unfortunately, their data lacks novelty and several reports on positive effects of IL6 in spinal cord injury described by several groups were simply not mentioned (Mukaino et al. *Exp Neurol* 2020; Yang et al., *Int Journal of Mol Med*, 2017; Yang et al *Exp. Neurol* 2012, Cafferty et al *JNeuro*, 2004). Thus it is not entirely surprising given the works of others and work by the authors in the optic system that IL6 stimulates axon regeneration. Furthermore, I have several concerns both with data interpretation and the robustness of data as detailed below. Therefore I cannot support publication and since a lot of work would be needed to justify claims made I would recommend rejection of the manuscript as it stands.

Major points:

Abstract/Introduction

- 1.) It is not clear why authors stress the "preinjury" timepoint for deletion of PTEN in PTEN ko (any reason?)
- 2.) It is not entirely true that no functional recovery was reported after PTEN inhibition: see Ohtake et al *Biomaterials* 2014; Danilov and Steward *Exp. Neurology* 2015

Results

- 3.) There is a conflict with existing data: In previous reports PTEN/SOCS3 mutants show synergistic effects suggesting cooperation of PTEN and SOCS3 (Stat3) signaling in regeneration in contrast to their study
- 4.) Given that IL6 is a central component of an immune response. Why did the authors never look at immune responses in the spinal cord?
- 5.) The effects on axon regeneration assessed with the tracer are not so strong as claimed and there are major fluctuations between mice when individual mice are plotted (Figure S3a, c). Effects at 6 or 7 mm are mainly due to 2-3 mice, the rest does not show major effects. Therefore, it is compulsory to show all animals as done by the authors and not just mean values.
- 6.) Functional axon regeneration was quantified with BMS score only which is not very

quantitative/objective and very prone to personal evaluation (authors did actually not state that BMS was performed blind to the cohorts). Further tests that allow clear quantification should have been done (ladder walk, rotarod or very sophisticated tests performed in studies by the Courtine group which allow for unbiased quantification)

7.) The expression of GP130 on regenerating neurons (e.g. raphe neurons) was not tested.

8.) Results, p5: authors mention neurons with an "atypical morphology" after hIL6 injection without further analysis and quantification. What should this indicate?

9.) The postinjury timepoint of hIL-6 injection is not specified precisely neither in results nor in materials methods. Was it injected 1h or 1d after SCC?

10.) Tracer injection: 2 weeks for tracer transport appear very long and authors should state whether this is the standard time window in the field.

11.) Results, p6: Authors mention "axonal dieback". This is not visible at all on this pictures and higher magnifications e.g. showing retraction bulbs have to be presented.

12.) Localization of hIL6 along on regenerating axons particularly of 5-HT neurons is not provided. Authors use GFP as a surrogate and only provide one example for axonal IL6 distribution in Fig. S1k

13.) Results, p8: how region specific is the DHT injection? Since it was done intracerebroventricularly I assume other brain regions might have also been damaged. No proof provided.

14.) Fig. S6: why did the authors use BV for the cell culture experiments and not the virus they used in vivo? Explanation is missing and results may be different when using the AAV-IL6 virus;

Fig. S6e: there must be a mistake in the scale otherwise average neurite length is only 6-8 um

15.) General comment: it is not always convincing that P-STAT3 staining is localized to neurons - more costaining should have been provided

16.) Results, p11: The Rosa-tdTomato approach is not convincing and solid quantification is missing; From Fig. 6i, it looks like that only a maximum of 10% of cells receive the RFP. This contrasts the approx. 50% of P-STAT3/5HT double positive cells (p10 bottom). So how relevant can this connection between cortical and raphe neurons actually be for the regeneration effects observed?

17.) Fig. 5b: cell loss should have been quantified

18.) Fig. 4a Pten/hIL6: why do the 5HT axons run in the middle of the spinal cord (in contrast to hIL6 alone and Fig. S7h)?

19.) Fig. 1k: total AKT and total STAT3 levels are missing

Discussion

20.) P12 Authors claim to show "direct evidence" for a cortical-raphe synapse; I do not see this and many other techniques would be required to firmly state this e.g. electron microscopy. Authors did not use any measures to show neuronal activation of 5HT neurons e.g. c-fos labeling would be appropriate.

21.) Since the authors can target the Red nucleus why did they not attempt to inject the hIL-6 virus directly into the Raphe nucleus to show more convincingly improved regeneration of this tract by IL6.

Methods

22.) Mouse strains: for how many generations have back-crossings been made or are animals still mixed of C57BL/6 and 129/Ola?

23.) Mice are overall very young (i.e. 4 month when sacrificing) - this should be mentioned more clearly and related to axon regeneration potential of older mice

24.) DHT Injection: was desipramine also injected in the control group?

25.) Quantification of Tracer positive axons: The ROIs the authors analyzed do not really allow for 3D mapping of axons and lightsheet approaches should have been used in order to clearly follow individual axons over 3D in the spinal cord

26.) BMS: authors should state whether observers were blind to cohorts. Also since this was done in an OF arena. How high was the arena? Did the observers have to judge the motor performance

of animals from above which would be prone to mistakes?

Minor Points:

27.) In the text authors label figures with capital letters whereas in figures small form letters are used.

28.) Authors use very frequently "strikingly", and "remarkably" to describe their findings which is not always appropriate

29.) P9: in one Figure it says "Fig. 54"

30.) Figure S1b. Why were GFAP cells also labelled in blue?

31.) Figure S4b, d and Fig. 5f: some dots are in between timepoints and it is not clear to which one they belong.

Reviewer #3:

Remarks to the Author:

The authors investigate the effect of hyper-il-6 on regeneration after spinal cord injury, and show that a single AAV injection of hIL-6 can promote regeneration. They discover that serotonergic neurons also regenerate due to the transneuronal transport of hIL-6 from CST axons to their terminal connections in the Raphe Nucleus. hIL-6 has previously been shown to increase neutrophils and macrophages when administered directly into the lesion site after SCI and thus have inhibitory effects. However, the current group has previously shown that hIL-6 is effective in promoting RGC regeneration. But this is the first time, the effect of hIL-6 has been shown to improve regeneration after SCI. Overall, this study is well done and uncovers an interesting link that will promote the recovery of two major descending pathways, CST and serotonergic axons, through the transfer of hIL-6 transsynaptically. The results from this study will be useful in thinking about major secondary effects that contribute to regeneration and raise a valid point about the cascading effect of treatments that shouldn't be ignored.

Major comments:

1. The abolishment of functional recovery after administering DHT is apparent in the BMS score, but the mechanism through which DHT might be abolishing this effect is unclear. If 5HT neurons are abolished by DHT after the improvement seen at 6w, is the functional recovery lost as a result of degenerating axons? If so, is there a visible defect within one day?? If this is the case, images of the spinal cord, stained for 5HT, similar to Fig 4 should be shown. Also whether the CST axons are affected by DHT treatment should be shown/addressed.

2. How is the Raphe nucleus being defined anatomically? As the authors have mentioned, the raphe nucleus extends rostrocaudally; is a specific region chosen?

3. Figure 6 - Transneuronal transduction of hIL-6 is the highlight of the paper. Therefore this warrants a more thorough quantitation. For example, the number of transduced cells (yellow cells) should be quantified in the RN? It appears that NRM doesn't have any transduced cells; is there a regional preference since the raphe spans a modest distance rostrocaudally?

Additionally, since the projections of cortical motoneurons into the raphe are being described here for the first time, other approaches to cross-verify these projections should be utilized.

Retroviruses is a simple example of one approach.

4. In 6b-d, the green background is very high, the colocalization is barely distinguishable. It is really not clear what one is looking at. Green axons in the RN are visible, but the yellow, blue and red arrows are confusing and have not been explained. Panel B seems unnecessary.

5. Supp Fig 1- fewer arrows in b-e. Panel F should be quantified and IL6 staining should be shown

in GFP control group for comparison. This is absolutely necessary to determine if IL6 is in fact increased in the hIL6 group.

Minor Comments:

1. The authors mention that no BDA-labelled axons were seen past 11mm and that confirms the completeness of the injury. This sounds like a good measure, but should be included in the supplemental figure.6. Panels like D should try and include DAPI or have a higher gain so the tissue is visible.
2. Figure 7G shows RN as the red nuclei, but previous figures refer to Raphe Nucleus as RN. Please be consistent.
3. Figure 2i has huge error bars which is concerning especially for the PTEN^{-/-} group. Can this be explained?
4. Figure 4- the authors inject hIL-6 bilaterally but don't see any improvement in regeneration, which is counter-intuitive. The authors should mention relevant information about the percentage of ipsi and contralateral projections for these tracts, which will be useful for readers in thinking about these results and explain the rationale behind why a bilateral injection was tested.
5. Figure 5F should include a legend to indicate which group is in red and black. The statistical significance is shown within each group; however the statistical significance between gfp and hIL-6 groups is also important and should be shown or mentioned in the figure legend at the very least, especially for 6w1d time point?
6. Supp Fig7H is unclear. Is this a horizontal section??
7. The methods section has a paragraph describing Intraspinal injections of hIL-6. Perhaps I missed it, but I didn't see intraspinal injections being described in any of the figures or main text. Please check thoroughly.

Reviewers' comments:

Reviewer #1: This is a comprehensive set of studies showing remarkable effects of AAV-hIL-6 injections into the sensorimotor cortex on regeneration of axons below a crush injury site. Functional recovery was shown to be dependent on regeneration of serotonergic raphespinal tracts, while release of hIL from axon connections between motor cortex and serotonergic raphe nuclei was demonstrated. Overall, this is an outstanding innovative study that reveals novel information that could potentially influence thinking in the field and contribute to the development of treatment options for spinal cord injury.

Response: We appreciate this comment.

Reviewer #1: Main concerns are related to practices and reporting of rigor and allowing for complete reproducibility. 1. Many of the experiments include only low numbers of animals, with sample sizes of 3, 4, or 5. In addition, there is a lack of adequate reporting of the outcomes of the statistical analyses. Since underpowered studies are detrimental to the ability to translate or build on findings from preclinical studies, it is important that rigor of the studies is clearly demonstrated. These are the recommendations:

- Authors need to comment on the rigor of their studies in light of the low sample sizes. For example: Were power analyses conducted? Were key controls included or findings replicated from previous studies?

Response: Concerning replicated findings from previous studies: As stated in the manuscript, we reproduced data on CST regeneration by PTEN^{-/-} from previous studies: Liu et., 2010; Geoffroy et al. 2015; Zukor et al., 2013.

We also agree that the rigor of our study should be as high as possible. We, therefore, performed additional repeating experiments so that all sample sizes are now n=6-10, as detailed in the table below. Furthermore, we investigated the main parameter (functional recovery after severe spinal cord crush) for the main groups (AAV-GFP vs. AAV-hIL-6) in several experiments presented in different figures (Fig. 3, Fig. 5, Fig. S7, Fig. S9). Thus we analyzed in total 29 AAV-GFP animals vs. 39 AAV-hIL-6 animals in the same experimental paradigm under very similar conditions (blinded evaluation, same surgent, etc.). Thus the effect is highly reproducible. To make this clear, we added the following text in the M&M section (page 32): "Among all studies concerning anatomical axon regeneration and functional recovery, we analyzed in sum 29 AAV-2 GFP and 39 AAV-2 hIL-6 treated wild-type animals comprising male (gfp, n=14; hIL-6, n=14) and female (gfp, n=15; hIL-6, n=25) animals. No gender-specific differences were observed in any of the analyses."

Figure	old n=	revision n=	Figure	old n=	revision n=
Figure 1 n-p			Figure 4j		
gfp 1w	3	6	gfp	5	7
hil6 1w	4	7	PTEN ko	6	6
gfp 3w	4	10	hil-6	9	9
hil6 3w	4	10	PTEN ko+ hil	10	10
gfp 5w	4	7	Figure 5f		
hil6 5w	4	7	gfp	5	6
gfp 8w	3	6	hil-6	6	6
hil6 8w	3	6			
pten	3	6	Figure 7c		
Figure 1 r, s			gfp	3	7
gfp	4	6	hil-6	3	8
hil-6	4	6			
PTEN ko	4	6	Figure S2 b, i, l, m		
Figure 2 C, l			gfp	5	7
gfp	5	7	PTEN ko	6	6
PTEN ko	6	6	hil-6	9	9
hil-6	9	9	PTEN ko+ hil	10	10
PTEN ko+ hil	10	10	Figure S6 b		
Figure 3 b-d			gfp	2	6
gfp	5	7	hil-6	2	6
PTEN ko	6	6			
hil-6	9	9			
PTEN ko+ hil	10	10			

Reviewer #1: Furthermore, authors need to provide all statistical details for all analyses including F values, dfs, and exact p values for all comparisons in supplementary tables. In several of the figures, variability of the data is so large that is difficult to comprehend that data passed tests for normality and weighted posthoc tests for further pairwise comparisons.

Response: We followed this suggestion. All statistical data are now presented in detail in a supplementary Table 1. Additionally, exact p-values are now directly included in the figures.

Reviewer #1: Currently, authors differentiate between different p-values with separate symbols. However, there is no physiological relevance to differences in p values; rather lower p values usually suggest less variability in the data. It is therefore not informative nor encouraged to have different indicators for different p values in the figures and one indicator is sufficient.

Response: We followed this suggestion. The revised manuscript presents the exact p-values instead of different indicators.

Reviewer #1: Authors need to provide detailed information for practices that were used to eliminate bias in all analyses.

Response: As stated in the paper, the experimenters were blinded to all treatments to avoid bias. This is now mentioned in the respective sections:

- **page 24 "...Cultures were arranged in a pseudo-randomized manner on the plates so that the investigator would not be aware of their identity."**
- **page 32/33 "...For all analysis including BMS testing, CST regeneration/sprouting/dieback, RpST regeneration, characterization of DHT treatment and immunohistochemistry individual mice or samples of respective animals were randomly labeled with different letters, in a way that the investigator was blinded to the identity of cohorts..."**

If there is additional information required in the text, we kindly ask for specific advice.

Reviewer #1: Authors need to expand on details in the methods section to allow for full and complete reproducibility. Below are a few examples, but authors should expand on many parts of the methods to provide all details needed for other groups to reproduce these experiments. - Western blot analysis; how were data analyzed over the technical replicates?

Response: We followed this suggestion and added the requested information to the M&M section (page 26): "Western blots were repeated using lysates from different animals as independent biological replicates. Each blot contained samples of a control group, which was used as a reference for data normalization. For this purpose, control samples were either loaded individually or mixed in equal proportions. Pixel intensities of bands were densitometrically quantified using ImageJ software and normalized to intensities of the respective loading control. Moreover, in each replicate treatment groups were normalized to the relative intensity of the control group. Individual blots used for quantification are shown in the new Fig. S14, and Fig. S15".

Additionally, all individual blots used for quantification are now included as supplementary Fig. S14 and S15.

Reviewer #1: CST sprouting: what are the axon density index and axon index? Explain more clearly in methods how these were calculated and explain when first mentioned in figure legends. What exactly is displayed on the y-axes in Figure 2? it is hard to imagine that the few axons visible in Figure 2e-h divided by BDA labeled fibers in the medulla can results in the numbers shown in the graph in 2i. Please clarify.

Response: The retraction of axons rostral to the injury site was quantified as an axon density index. This value was determined by measuring the BDA staining intensity as described by others previously (Geoffroy et al., 2015; Liu et al., 2010; Zukor et al., 2013).

To address this issue, we added the following text into the M&M section (page 28): “A series of 100 μm wide rectangles covering the entire dorsoventral axis of the spinal cord were superimposed onto sagittal sections, starting from 1.5 mm rostral up until the injury site, which was visible by its morphology and additional GFAP staining of the glial scar. The mean gray value of the background signal in BDA negative areas was measured in each spinal cord section and subtracted later from quantified BDA staining. For BDA staining quantification, the mean gray values of each rectangle were measured at every 100 μm for 1.5 mm to 0 mm rostral to the lesion site. After subtraction of the background values were divided by the mean gray value at 1.5 mm rostral to the lesion site. At this distance, BDA staining intensity represents the total amount of labeled axons in the main CST since retraction is usually limited to a distance of ~500-1000 μm proximal to the lesion site (Liu et al., 2010; Zukor et al., 2013; Geoffroy et al., 2015). By quantifying and averaging all consecutive sagittal sections through the spinal cord containing the main CST from each animal, we received the axon density index, which was then plotted as a function of the distance to the injury (-1.5 mm - 0 mm).”

Also, the axon index in Fig. 2 I shows the total number of BDA labeled regenerated CST axons quantified over all consecutive sections of each spinal cord. This value was then divided by the total number of BDA labeled axons in the medulla, as displayed in supplementary figure 2b. We agree with the reviewer that there was a typo on the y-axis, which is corrected in the revised version of the manuscript. The numbers should be seen as multiplied by 10^{-3} instead of 10^3 .

To describe the axon index more precisely, we added the following sentences into the methods part (page 28/29): “A grid with lines spaced every 100 μm was aligned to each image, and the number of axons at 1 mm, 5 mm, 6 mm, and 8 mm from the caudal edge of the lesion were quantified in ~40 adjacent sections per animal. For quantification, all axons observed within an area of 0.5 mm caudal and rostral to the respective measuring point were counted. To obtain the axon index, the total number of BDA labeled regenerated CST axons was quantified at each distance over all consecutive sections of each spinal cord and then divided by the total number of BDA labeled axons in the medulla as described above and displayed in Fig. S2 b”

Reviewer #1: Evaluation of CST regeneration: it is mentioned that only animals with complete lesions were included in the evaluation. What portions of animals were excluded and what were the final sample sizes? Since the authors presumably have the behavioral data for animals without complete lesions, it may be useful to also provide the behavioral recovery of these incomplete lesions, which will add to the discussion of comparisons with other less severe injury models.

Response: Initially, 10 animals for each group were subjected to surgery and treatment for the study. The final sample sizes are given in the respective figure legends (PTEN^{+/+}/gfp, n=7; PTEN^{+/+}/hIL-6, n=9; PTEN^{-/-}/gfp, n=6; PTEN^{-/-}/hIL-6, n=10). Although the crush injury itself was relatively easy to perform and highly reproducible, some animals died during or after surgery (bladder problems, etc.). Thus, only very few animals were excluded because of spared axons: These were only two animals with PTEN^{-/-} and one control mouse that had received AAV-GFP. The two PTEN^{-/-} animals had a final BMS score of 5 and 3, respectively. The control animal showed a final BMS score of 3.

We believe that including these data is not helpful for the readers and somewhat distracting. This is because without having any hIL-6 treated animal with spared axons and the low numbers, no comparison is possible. Additionally, the proportion and localization of spared axons were different in these 3 animals making a comparison with less severe injury models such as dorsal hemisection difficult. However, if the editors feel that these data should be included in this study, we could add these as supplementary figure 16.

Reviewer #1: Male and female mice were used: provide exact numbers for each study and were there sex differences in any of the analyses?

Response: Male and female mice were randomly distributed among the treatment groups for all experiments. We did not observe any sex differences in any of the analyses, which is now stated in the text. Exact numbers for respective studies, including final BMS scores of respective animals, are given in the following table.

Fig. 2; 3B; 4; S2; S4; S5; S6				Fig. 3F				Fig. S7			
genotype	animal	BMS 8w	gender	genotype	animal	BMS 8w	gender	genotype	animal	BMS 8w	gender
wt	gfp 1	2	F	wt	HIL6 1	4	F	wt	gfp 1	1	F
	gfp 2	2	F		HIL6 2	3.5	F		gfp 2	0	F
	gfp 3	2.5	M		HIL6 3	4.5	M		gfp 3	2	F
	gfp 4	1.5	M		HIL6 4	3.5	M		gfp 4	1.5	F
	gfp 5	1	M		HIL6 5	2.5	F		gfp 5	2	M
	gfp 6	2	F		HIL6 6	4	F		gfp 6	1.5	M
	gfp 7	1.5	M		HIL6 7	2.5	M		gfp 7	2	F
pten ko	gfp 1	1.5	M		HIL6 8	4	F		gfp 8	1	F
	gfp 2	2	M		HIL6 9	5	F		HIL6 1	5	F
	gfp 3	2.5	M				HIL6 2		4	F	
	gfp 4	2	M				HIL6 3		5	F	
	gfp 5	3	F				HIL6 4		3.5	M	
	gfp 6	1.5	F				HIL6 5		2.5	M	
wt	HIL6 1	4	F	Fig. 5							
	HIL6 2	3	F	genotype	animal	BMS 6w	gender				
	HIL6 3	2.5	M	wt	gfp 1	2.5	M	Fig. S9			
	HIL6 4	3.5	M		gfp 2	2	M	genotype	animal	BMS 8w	gender
	HIL6 5	5	M		gfp 3	2.5	M	bl6 wt	gfp 1	2	M
	HIL6 6	4	F		gfp 4	2	M		gfp 2	3	F
	HIL6 7	4	F		gfp 5	2.5	M		gfp 3	2	F
	HIL6 8	4	F		gfp 6	2	F		gfp 4	2	F
	HIL6 9	3.5	M		HIL6 1	3	F		gfp 5	2	F
pten ko	HIL6 1	5	F		HIL6 2	3	F		gfp 6	3	F
	HIL6 2	2.5	F		HIL6 3	3	M		gfp 7	2	M
	HIL6 3	4	F		HIL6 4	4	M		gfp 8	2	M
	HIL6 4	5	M		HIL6 5	4	F		HIL6 1	5	F
	HIL6 5	3	M	HIL6 6	5	F	HIL6 2		2	F	
	HIL6 6	5	M				HIL6 3		5	M	
	HIL6 7	3.5	M				HIL6 4		5	F	
	HIL6 8	7	M				HIL6 5		5	F	
	HIL6 9	5	M				HIL6 6		3	F	
	HIL6 10	5	M				HIL6 7	2	F		

To provide an overview regarding functional recovery, we summarized all 29 GFP and 39 hIL-6 treated wild type animals among all studies and show BMS scores for male (gfp n= 14, hIL-6 n=14) and female (gfp n=15, hIL-6 n=25) cohorts separately as well as total values of both sexes in the following graph:

We added the following sentences into the M&M section (page 32): “Male and female mice were randomly distributed among the treatment groups for all experiments. Among all studies concerning anatomical axon regeneration and functional recovery, we analyzed in sum 29 AAV-2 GFP, and 39 AAV-2 hIL-6 treated wild type animals containing male (gfp n= 14, hIL-6 n=14) and female (gfp n=15, hIL-6 n=25) animals. No gender-specific differences were observed in any of the analyses.” If deemed necessary, we could include the diagram above as another suppl. Fig., but honestly feel that this is not more informative.

Reviewer #1: Immunohistochemistry: what validation procedures were used for the primary antibodies and for staining protocols?

Response: All antibodies used in the current study were validated by the vendor and used/validated in many previous studies, which are listed in detail in the nature

reporting summary PDF. Additionally, the staining with the pSTAT3 antibody showed typical nuclear localization and was validated in our previous studies, for example, by the absence of the staining in conditional STAT3^{-/-} mice after inflammatory stimulation (Leibinger et al., 2013, Fig. 1). Regarding hyper-IL-6 staining, the antibody showed only a signal in cell bodies and axons of AAV-hIL-6 transduced, GFP positive cells but not in AAV-GFP treated control animals. This was also validated by overexpression in retinal ganglion cells or HEK2903 cells in our previous publication (Leibinger et al. 2016). The pS6 staining was only induced in PTEN^{-/-} cells. All these details are also listed in the nature reporting summary PDF.

Reviewer #1: A large amount of immunostaining was performed for labeled cells (for example pStat3), but it is unclear if any quantitative analyses were performed and thus if the images shown in the figures are representative of the data. Since western blot analyses were conducted with n=3 sample sizes, adding quantitative analysis of numbers of immunoreactive cells would have greatly strengthened the findings. Confocal image capturing and use of standardized camera settings was not described for any analyses.

Response: To address this suggestion, we performed additional experiments to increase the sample size of the Western blot data to n=6-10 of individual biological samples. Respective data are presented in Fig. 1 (n=6-10) and in Fig. 6 b,c (n=7-8). Moreover, Western blots of the same samples show that, in contrast to phospho-STAT3, total STAT3 protein expression is not significantly changed by AAV-hIL-6 treatment. Additionally, we provide an overview through 100 micrometers of transverse dissected cleared tissue, showing a large proportion of pSTAT3 positive cells around the injection sites (Fig. S1 k). A similar scan through cleared tissue is also provided for the brainstem (Fig. S12 b.), showing that pSTAT3 is activated over the whole rostrocaudal length of the raphe nuclei. Finally, we performed surgery on additional animals and generated immunohistochemical sections to provide new photos in higher resolution showing pSTAT3 positive neurons in the cortex (Fig. 1 b-f) and medulla (Fig. 6 d-u) following cortical AAV-hIL-6 treatment.

To provide the requested information about image capturing, we added the following sentences into the M&M part (page 32): “For all images that were taken with the widefield fluorescence microscope (Axio Observer.D1, Axio Scan Z1, Zeiss, VS120, Olympus), we used standardized camera settings and the same exposure times among all samples. For confocal imaging (LSM 510, Zeiss; SP8, Leica), standardized settings of the resolution, z-stacks, pinhole, laser intensity, detector gain/offset were used for all comparatively captured images among all samples.”

Reviewer #1: Figure 7 (and in several other figures): it is very difficult to see dual labeling. Perhaps show separate images or use arrows.

Response: To address this issue, we added photos of higher resolution and better quality so that the colocalization appears now more apparent. These are now included in Fig. 6 d-u.

Reviewer #1: Panel I, Figure 7: it appears that there are mainly BDA labeled cells and only very few axons in the red nucleus of the hIL-6-treated animal. The BDA labeling appears to be very different compared to control; is that a real observation across all animals? The images in this figure don't appear to support the finding that pSTAT3 labeled neurons are targeted by BDA fibers.

Response: This comment is likely based on a misunderstanding. Only panel h shows BDA labeled CST axons that are targeting the red nucleus. This photo visibly shows a large number of projecting axons. This projection is known and was also described in several previous studies (Z'Graggen et al., 2000; Joeng et al., 2016). In contrast, in the panel I, BDA was injected directly into the red nucleus (RedN) using a stereotactic frame and coordinates described in the M&M section to label RedN neurons and not CST axons. This is why, mainly cells but not axons are visible. To make this point better understandable, we added the following sentence in the respective figure legend (new Figure S12 c-e)“... RedN cell bodies were visualized by BDA (green) after

direct injection into the red nucleus using specific coordinates 2 weeks before tissue harvest...”. The supposed difference between control and AAV-hil-6 treated samples does not represent a real observation.

Reviewer #1: Supplemental materials: Figures and videos were very informative, except for video 7: the red signal isn't visible and it is unclear without any labeling of landmarks exactly where anatomically this video is rendered.

Response: Due to limitations in the respective software, we were not able to include sufficient annotations. But the video represents a three-dimensional rendering, which is based on the same picture presented as a projection now in figure S12 b. This information and the illustration in Fig. S12 should help with orientation. To prevent potential misunderstandings, this is now clearly linked in the Video legend on page 54 “... localization of the imaged area and a 2D projection image, including a scale bar, are presented in Fig. S12 A, B...”.

Reviewer #2: In the manuscript by Leibinger et al., the authors use a previously described hIL6 viral application approach tested in optic nerve regeneration in spinal cord injury. They observe improved numbers of regenerating axons, particularly raphespinal axons and elevated BMS scores. They argue that a transneuronal delivery of IL6 from cortical to raphe neurons is the underlying mechanism and emphasize the advantages of their approach over regeneration effects observed in PTEN mouse mutants.

Overall, the authors show some effects on regeneration and the transneuronal mechanism might be very interesting. Unfortunately, their data lacks novelty and several reports on positive effects of IL6 in spinal cord injury described by several groups were simply not mentioned (Mukai et al. *Exp Neurol* 2020; Yang et al., *Int Journal of Mol Med*, 2017; Yang et al. *Exp. Neurol* 2012, Cafferty et al. *JNeuro*, 2004). Thus it is not entirely surprising given the works of others and work by the authors in the optic system that IL6 stimulates axon regeneration. Furthermore, I have several concerns both with data interpretation and the robustness of data as detailed below. Therefore I cannot support publication and since a lot of work would be needed to justify claims made I would recommend rejection of the manuscript as it stands.

Response: As detailed in our responses below, this reviewer's comments appear to be mostly based on misunderstandings. Thus, some comments are unclear.

Reviewer #2: It is not clear why authors stress the "preinjury" timepoint for deletion of PTEN in PTEN ko (any reason?)

Response: We mention this because the deletion of PTEN was induced before the injury, while the AAV hIL-6 treatment started right after the injury. From a therapeutical point of view, this is highly relevant.

Reviewer #2: It is not entirely true that no functional recovery was reported after PTEN inhibition: see Ohtake et al. *Biomaterials* 2014; Danilov and Steward *Exp. Neurology* 2015

Response: This is not what our paper said, and we don't see any conflict with the data shown in these publications. Others also failed to see functional recovery after a complete spinal cord crush and PTEN^{-/-}. In contrast, Danilov and Steward used mice that had received moderate contusion injuries at cervical level 5, and Ohtake et al. used dorsal hemisection. As clearly pointed out in our discussion, these are much less severe injury models than the one used in our study, which is also reflected by the BMS data in these studies. For example, in Ohtake et al., even control animals reach a score of ~6 3 weeks after injury. We, therefore, do not understand this comment, as this point was clearly stated in the discussion of our manuscript (page 13): "In contrast to incomplete, less severe hemisection- or contusion-injury models, permitting spontaneous functional recovery with BMS scores ≥ 6 ^{11,21,23}, the SCC, just as in a complete transection, eliminates all axonal connections between the proximal and distal portions of the spinal cord. This injury model is therefore highly resistant towards functional recovery and shows only some active flexion movements of hindlimbs (BMS score of ≤ 2)^{10,21-23,33}. Hence, most previous treatment strategies in this injury model did not reach significant effects beyond a reduction of dieback of CST axons³⁴⁻³⁶. Only PTEN^{-/-} in cortical motoneurons enabled the regeneration of CST-axons and synapse formation with interneurons but still failed to restore locomotion^{10,11,33,37,38}."

Reviewer #2: There is a conflict with existing data: In previous reports PTEN/SOCS3 mutants show synergistic effects suggesting cooperation of PTEN and SOCS3 (Stat3) signaling in regeneration in contrast to their study

Response: In our opinion, there is no conflict with the existing literature since we also saw synergistic effects of PTEN ko and hIL-6 in respect of CST axon regeneration (Fig. 2 i.) Our findings are in line with the existing literature and also with our previous publication that PTEN^{-/-} and hIL-6 enhance optic nerve regeneration synergistically (Leibinger et al., 2016). In contrast, the observation that we don't see synergistic effects on raphespinal tract regeneration is because PTEN^{-/-} via AAV2 remains restricted to the motor cortex and CST, while hIL-6 is transported and

transsynaptically released protein to the raphe nuclei (-> serotonergic fibers). Therefore, in our paradigm, raphe neurons were treated only with hIL-6 but no PTEN^{-/-}.

Reviewer#2: Given that IL6 is a central component of an immune response. Why did the authors never look at immune responses in the spinal cord?

Response: The distance between the spinal cord and motor cortex is relatively high, and hIL-6 is released via supraspinal axonal branches in specific brain regions. Thus, as shown in the paper, it is only present in defined areas. Consistently, we did not detect pSTAT3 positive cells in the spinal cord after intracortical AAV-hIL-6 treatment (data not shown). Moreover, in contrast to a previous study, which directly injected the hIL-6 protein into the lesion site in rats (Lacroix et al., 2002) leading to an immunresponse concomitant with an increased lesion area, we did not observe any change of the size of the injury site as documented in Fig. S2 k, l.

Nevertheless, to address this issue, we performed additional experiments and analyzed CD11b positive activated macrophages/microglia in the spinal cord of intracortical AAV-GFP or AAV-HIL-6 injected mice either after spinal cord crush or no further treatment. As a result of this, we didn't observe any difference between the two groups after the spinal cord crush and observed no activated CD11b positive cells after either treatment without crush. These data are presented in the new supplementary Fig. 3.

Reviewer#2: The effects on axon regeneration assessed with the tracer are not so strong as claimed and there are major fluctuations between mice when individual mice are plotted (Figure S3a, c). Effects at 6 or 7 mm are mainly due to 2-3 mice, the rest does not show major effects. Therefore, it is compulsory to show all animals as done by the authors and not just mean values.

Response: This comment is likely based on a misunderstanding as we showed data of all single animals in the supplementary Fig 4. The reason not showing them in the main referring figure is that this one already has many bars. Additional dots for every single animal would make the graph look overcrowded and confusing. So for the sake of the quality of our paper, we would like to leave it as it is. However, if the editors feel that we should include the dots into the main figure, we could easily do so.

Reviewer#2: Functional axon regeneration was quantified with BMS score only which is not very quantitative/objective and very prone to personal evaluation (authors did actually not state that BMS was performed blind to the cohorts). Further tests that allow clear quantification should have been done (ladder walk, rotarod or very sophisticated tests performed in studies by the Courtine group which allow for unbiased quantification)

Response: As stated in the M&M part, BMS testers were always blinded to the identity of the cohorts. The same is true for the experimenters who evaluated anatomical regeneration. Thus, data were objective. Additionally, our videos prove that the effects on functional recovery are strong and unambiguous.

Nevertheless, we agree that the automatic evaluation of testing might be even more objective and convincing. Therefore, we bought a catwalk gait analysis system which is commonly used in the field since many years (Hamers et al., 2006; Steijger et al., 2013; VanMeteren et al., 2003; Lee et al., 2010; Koopmans et al., 2009; Floriddia et al., 2010; Crowley et al. 2018) and subjected more AAV-GFP or AAV-hIL-6 treated mice to surgery. The BMS scores of these animals were again, blinded for the examiners, determined, and these animals were analyzed in the catwalk system afterward. We used a ranking of catwalk performance (from 1=worst performing to 6=best performing) of all tested hil6 treated mice based on all evaluated catwalk parameters. This ranking was plotted against the BMS value of the individual animals and showed an almost exact linear correspondence between catwalk and BMS results, similar to previously shown by other studies (Crowley et al., 2018). These data, presented in the new supplementary figure 7.

Reviewer#2: The expression of GP130 on regenerating neurons (e.g. raphe neurons) was not tested.

Response: Others have already shown GP130 expression in raphe neurons (e.g., Watanabe et al. 1996; Rognum et al., 2009). Moreover, it has been shown by many labs that hIL-6 activates STAT3 via gp130 and we show a clear hIL-6-induced pSTAT3 induction in western blots with additional biological replicates (n=8). Also, we generated additional immunohistochemical data, which were included in Fig. 6 d-u. These new data confirm pSTAT3 in raphe neurons. Given the previously published evidence of gp130 expression in raphe neurons and the fact that hyper-IL-6 can activate STAT3 via gp130, gp130 stainings are redundant data and not informative. Nevertheless, to avoid any confusion to readers, we now included the respective references in the text of the manuscript (page 11): "...We then investigated whether raphe nuclei in the brain stem receive synaptic input from transduced cortical motoneurons, since raphe neurons are described to express the hIL-6 receptor GP130 (Rognum et al., 2009; Watanabe et al., 1996) and could also be stimulated by the transneuronal release of the designer cytokine...".

Reviewer#2: The postinjury timepoint of hIL-6 injection is not specified precisely neither in results nor in materials methods. Was it injected 1h or 1d after SCC?

Response: hIL-6 was injected 30 min after injury. This information is now detailed in the text (page 19).

Reviewer#2: Tracer injection: 2 weeks for tracer transport appear very long and authors should state whether this is the standard time window in the field.

Response: This is standard in the field, as shown in many previous publications (e.g., Liu et al., 2010, Zukor et al., 2013, Geoffroy et al., 2015; Joshi et al., 2015). To clarify this issue, we added the respective citations to the M+M section (page 21).

Reviewer#2: Results, p6: Authors mention "axonal dieback". This is not visible at all on this pictures and higher magnifications e.g. showing retraction bulbs have to be presented.

Response: The CST axon dieback following spinal cord injury is not a new finding by our study. Instead, it is known for decades and has already been well characterized in many previous studies, which were also cited in our manuscript. Nevertheless, we accept the reviewer's criticism of low resolution in our images and present pictures of higher magnification and resolution showing retraction bulbs in supplementary Fig. 2 n-r.

Reviewer#2: Localization of hIL6 along on regenerating axons particularly of 5-HT neurons is not provided. Authors use GFP as a surrogate and only provide one example for axonal hIL6 distribution in Fig. S1k

Response: This comment is based on a misunderstanding. 5-HT neurons should not show any h-IL6 since these neurons do not produce or take up hIL-6! Instead, the designer cytokine is only expressed in cortical neurons, released by CST axon terminals and then binds on extracellular domains of the receptor on the cell membrane of raphe neurons. This then leads to intracellular STAT3 phosphorylation. Therefore we showed a GFP signal in CST axons as a surrogate and demonstrated STAT3 phosphorylation in raphe neurons, which is additionally supported by new data: Western blot (n=8) (Fig. 6 b, c), immunohistochemical sections (Fig. 6 D-U) and by scans through cleared brain stem tissue (Fig. S12 b). In contrast, no pSTAT3 signal was detected in raphe nuclei of AAV-GFP treated controls. Moreover, we added new data showing hIL-6 protein expression in transduced cortical neurons in immunohistochemistry (Fig. S1g) and western blot (n=7) (Fig S1 h, i), which was also not detected after AAV-GFP control (n=8) transduction.

Reviewer#2: Results, p8: how region specific is the DHT injection? Since it was done intracerebroventricularly I assume other brain regions might have also been damaged. No proof provided.

Response: All serotonergic neurons were affected, but this has only little consequences for uninjured animals, indicating that other supraspinal neurons of descending tracts remain intact. As proof, we already provided data in Fig. 5 g and h. Moreover, we confirmed that cortical motoneurons are still intact and can actively transport BDA, which was injected into the cortex 1 week after DHT application (figure 5b). Additionally, we show in the revised figures that surrounding neuN-positive neurons in the brainstem are still present after DHT treatment (new Figure S7 k), while 5-HT positive neurons are almost entirely abolished (new supplementary Fig. 7 a-i). Moreover, DHT is an established standard treatment for exclusively depleting serotonergic neurons, and the specificity was already described in numerous previous studies (Fuxe et al., 1978; Benton et al., 1997; Sullivan et al., 2000; Cummings et al., 2009). To make this clearer, we added the respective citations into the manuscript on page 9.

Reviewer#2: Fig. S6: why did the authors use BV for the cell culture experiments and not the virus they used in vivo? Explanation is missing and results may be different when using the AAV-II6 virus; Fig. S6e: there must be a mistake in the scale otherwise average neurite length is only 6-8 um

Response:

- **BV: We used BV as AAV cannot be used in culture. This is because the expression takes too long for the neurons which survive only a few days in culture. As we described in detail (Leibinger et al., 2017; Levin et al., 2016), BV expresses protein rapidly and efficiently with a high transduction rate. Furthermore, we have shown before that hIL-6 can be well expressed in adult neurons (RGCs) via BV (Leibinger et al., 2016).**
- **Neurite growth of RGCs: The average neurite length per RGC was determined by dividing the sum of neurite length per well by the total number of RGCs per well (on average ~500 RGCs). The resulting neurite length per RGC also includes the majority of RGCs, which did not grow out any neurite (~90%), thereby representing neurite growth as well as neuritogenesis. Thus, obtained values are much lower than the average axon length of RGCs with extended neurites (~20–500 µm). We state this now more precisely in the M&M part on page 24. This evaluation method has been used in many of our previous and other papers (Heskamp et al., 2013; Leibinger et al., 2013; Leibinger et al., 2012; Leibinger et al., 2016; Leibinger et al., 2017; Leibinger et al., 2019; Levin et al., 2019). However, since we repeated this experiment several times more for the revision of this manuscript and to avoid confusion, we present the data now as normalized values relative to the control group of each experiment.**

Reviewer#2: General comment: it is not always convincing that P-STAT3 staining is localized to neurons - more costaining should have been provided

Response: To address this issue, we prepared additional animals and took new photos of better quality, now clearly confirming the localization of pSTAT3 in neurons (new Fig. 1 C-G. new Fig. 6 D-U).

Reviewer#2: Results, p11: The Rosa-tdTomato approach is not convincing and solid quantification is missing; From Fig. 6i, it looks like that only a maximum of 10% of cells receive the RFP. This contrasts the approx. 50% of P-STAT3/5HT double positive cells (p10 bottom). So how relevant can this connection between cortical and raphe neurons actually be for the regeneration effects observed?

Response: Also, this comment is likely based on a misunderstanding: Transneuronal transduction with AAV1-Cre can't be directly correlated to AAV2 mediated hIL-6 expression in cortical neurons and transneuronal delivery of the protein to raphe neurons. Despite the possibility of different transduction efficiencies of both viruses in cortical neurons, transneuronal transduction of AAV1 is a relatively rare event because AAV virus particles cannot be reproduced in the cell without specific helper plasmids. As a consequence, some AAV1 particles that are taken up by cortical motor neurons must stay intact and be directly transported along the axon to be released in different brain regions. Thus, not all synaptically connected neurons can receive transneuronal transduction. Therefore, this approach was only used as a proof of principle to show the raphe-cortical connection. The revised manuscript provides additional evidence for this connection by a retrograde AAV1 mediated approach (Fig. 7 h-u).

In contrast, hIL-6 (as a protein) is expressed continuously in high amounts (CMV promoter) (s. Fig. S1 g-i), transported along the axon, and released to raphe nuclei. Therefore, it is conceivable that the amount of pSTAT3 positive raphe neurons (~50%) is much higher than the transduction rate after AAV1-Cre mediated transneuronal transduction. Nevertheless, we observed transneuronally AAV1-Cre transduced serotonergic neurons over the whole rostrocaudal length of the raphe nuclei. For the reasons lined out above, quantification is not needed.

To clarify this point, we added the following statements to the discussion part on page 16: "... Moreover, we observed transneuronal transduction over the whole rostrocaudal length of the raphe nuclei, even though the overall transneuronal transduction rate was expectedly low. This is due to the specific properties of AAVs, which are not able to reproduce themselves without specific helper plasmids. This means virus particles have to remain intact after entering the cell and have to be directly transported along the axons for later synaptic release, which seems to be a rather rare event. ... In contrast to AAV1 mediated transneuronal transduction, hIL-6 is continuously high expressed, transported, and released at axon terminals to secondary neurons as reflected by a high percentage of pSTAT3 positive serotonergic neurons (~50%) in all brain stem raphe nuclei."

Reviewer#2: Fig. 5b: cell loss should have been quantified

Response: We followed this suggestion and quantified the loss of 5HT-positive neurons in n=7-8 animals 1 day after DHT treatment. Data show an evident effect in the raphe nuclei with an apparent loss of 5HT-positive axons in the spinal cord (new supplementary Fig. 8 a-j)

Reviewer#2: Fig. 4a Pten/hIL6: why do the 5HT axons run in the middle of the spinal cord (in contrast to hIL6 alone and Fig. S7h)?

Response: Fig. 4 a shows regenerating axons from PTEN^{-/-}/hIL-6, and merely hIL-6 treated mice, which are distributed over the whole width of the spinal cord, indicated by the white arrows. The number of regenerating axons in different areas varies, of course, between different sections. Panels: f, g, h, I to which this reviewer might refer to, just show selected magnifications of regions with axons. We agree that this could be explained better in the figure legend. To address this issue, we included the following text on page 37: "As typical for regenerating RpST axons, they were located over the whole dorsoventral width of the spinal cord. Examples of regenerating axons are indicated by dashed boxes and white arrows"

Reviewer#2: Fig. 1k: total AKT and total STAT3 levels are missing

Response: We partially followed this suggestion and measured total STAT3. These data are now included in Fig. 1 n, p, and Fig. 6 b,c. However, showing total AKT is not necessary as pAKT levels did not change by hIL-6.

Reviewer#2: Authors claim to show “direct evidence” for a cortical-raphé synapse; I do not see this and many other techniques would be required to firmly state this e.g. electron microscopy. Authors did not use any measures to show neuronal activation of 5HT neurons e.g. c-fos labeling, would be appropriate.

Response: To address this issue, we additionally performed retrogradely transsynaptic labeling (see below). Thus, several experiments provide sufficient evidence for a cortical-raphé connection.

- **New data: Retrograde transsynaptic labeling with AAV1 from raphe nuclei to the motor cortex (Fig. 7 G-L).**
- **Anterograde AAV1 mediated transsynaptic labeling from the motor cortex to the raphe nuclei (Fig. 7 A-F).**
- **STAT3 activation in serotonergic raphe neurons close to GFP-positive hIL-6 transduced cortical neurons (Fig. 7 D-U) also proven by Western blot from samples of 8 individual animals (Fig. 6 B, C)**
- **Anatomical regeneration of the raphe spinal tract after cortical AAV2 hIL-6 application but not after cortical AAV2 mediated PTEN ko (Fig. 4).**
- **It was already shown in previous studies that AAV1 is transferred via synapses to secondary neurons in an anterograde and retrograde manner (Jeong et al., 2016, Zingg et al., 2017; Tervo et al., 2016; Rothermel et al., 2013; Aschauer et al., 2013). Moreover, as cited in the manuscript also data from Jeong et al. point to a cortical-raphé connection.**

Moreover, we directly show the activation of regenerative signaling by pSTAT3 staining in 5HT-positive neurons (see above), which is relevant to our paper. We do not share the opinion that c-fos staining is more appropriate than pSTAT3.

Reviewer#2: Since the authors can target the Red nucleus why did they not attempt to inject the hIL-6 virus directly into the Raphe nucleus to show more convincingly improved regeneration of this tract by IL6.

Response: These experiments are beyond the scope of the current manuscript because studying the red nucleus alone would not be sufficient. We might also have to activate the other supraspinal nuclei as well directly. Addressing these questions is part of an extensive follow-up study.

Reviewer#2: Mouse strains: for how many generations have back-crossings been made or are animals still mixed of C57BL/6 and 129/Ola?

Response: Mice were not further backcrossed. This is now stated in the M&M part. However, based on our previous publications (Leibinger et al., 2016, 2017, 2019), these mice behave like 129/Ola animals regarding CNS regeneration and are less refractive than BL6 mice. This is why we tested an additional cohort of pure BL6 mice already in the original manuscript.

Reviewer#2: Mice are overall very young (i.e. 4 month when sacrificing) – this should be mentioned more clearly and related to axon regeneration potential of older mice

Response: Animals are generally considered an adult at the age of 7-8 weeks, and most other labs use animals of similar age (Liu et al., 2010; Geoffroy et al., 2015, Zukor et al., 2013). Moreover, we mention the age in the M&M section.

Reviewer#2: DHT Injection: was desipramine also injected in the control group?

Response: Yes, this is now clearly stated on page 21.

Reviewer#2: Quantification of Tracer positive axons: The ROIs the authors analyzed do not really allow for 3D mapping of axons and lightsheet approaches should have been used in order to clearly follow individual axons over 3D in the spinal cord

Response: The current study is already very comprehensive. The main focus of the current study with 7 main, 15 supplementary figures, and 8 videos lies in functional recovery and the transneuronal delivery mechanism of hIL-6. In terms of anatomical regeneration, we make the point that cortical hIL-6 application promotes CST and RpST regenerate. A detailed analysis of 3D mapping would be far beyond the scope of the paper and be another comprehensive follow up study. This is, in particular, true, because we show that CST regeneration does not substantially contribute to functional recovery and that next to the RpST also other transneuronally activated tracts (e.g., rubrospinal tract) could contribute to the restoration of motor function. Thus, we feel that this recommendation it is not necessary to support our findings.

Reviewer#2: BMS: authors should state whether observers were blind to cohorts. Also since this was done in an OF arena. How high was the arena? Did the observers have to judge the motor performance of animals from above, which would be prone to mistakes?

Response: During testing, the investigators were blinded to the identity of the cohorts. Animals from different treatment groups were randomly distributed between cages. This information is now given in the text.

The comment regarding the arena is not clear to us, as we also provide videos of the recovered animals. Nevertheless, to clarify this issue: The open-field arena was surrounded by a 30 cm high transparent acrylic glass plane, which prevented the mice from escape but still allowed observation from the side view (see image below).

Additionally, the side view is also shown in the supplementary videos.

Moreover, we now provide additional data of automated catwalk gait analysis for hIL-6 treated animals in the revised manuscript showing similar results as the BMS tests (supplementary Fig. 7 and video 5).

Reviewer#2: In the text authors label figures with capital letters whereas in figures small form letters are used.

Response: In the current version of the paper, all labels have been changed to small letters.

Reviewer#2: Authors use very frequently “strikingly”, and “remarkably” to describe their findings which is not always appropriate

Response: We carefully checked our paper for these words and removed them, where we felt they might be inappropriate. We ask for more detailed guidance if still judged inappropriate.

Reviewer#2: P9: in one figure it says “Fig. 54”

Response: Thank you for this comment. We corrected this typo in the current version of the manuscript.

Reviewer#2: Figure S1b. Why were GFAP cells also labelled in blue?

Response: GFAP signals were shown in black and white (b) for optimal contrast and additionally in a costaining with GFP (green). As GFP is usually green, we chose blue for GFAP to indicate the colocalization.

Reviewer#2: Figure S4b, d and Fig. 5f: some dots are in between timepoints and it is not clear to which one they belong.

Response: The single dots are already very close to each other. We would have to decrease the distance between the single dots. Changing this would cause a partial overlap, and some values of individual animals would not be visible anymore.

However, if the editors feel that this change is required, we are willing to make it, although we are afraid that most readers might not appreciate this.

Reviewer #3: The authors investigate the effect of hyper-il-6 on regeneration after spinal cord injury, and show that a single AAV injection of hIL-6 can promote regeneration. They discover that serotonergic neurons also regenerate due to the transneuronal transport of hIL-6 from CST axons to their terminal connections in the Raphe Nucleus. hIL-6 has previously been shown to increase neutrophils and macrophages when administered directly into the lesion site after SCI and thus have inhibitory effects. However, the current group has previously shown that hIL-6 is effective in promoting RGC regeneration. But this is the first time, the effect of hIL-6 has been shown to improve regeneration after SCI. Overall, this study is well done and uncovers an interesting link that will promote the recovery of two major descending pathways, CST and serotonergic axons, through the transfer of hIL-6 transsynaptically. The results from this study will be useful in thinking about major secondary effects that contribute to regeneration and raise a valid point about the cascading effect of treatments that shouldn't be ignored.

Response: We appreciate this comment.

Reviewer #3 If 5HT neurons are abolished by DHT after the improvement seen at 6w, is the functional recovery lost as a result of degenerating axons? If so, is there a visible defect within one day?? If this is the case, images of the spinal cord, stained for 5HT, similar to Fig 4 should be shown. Also whether the CST axons are affected by DHT treatment should be shown/addressed.

Response: To address this comment, we generated new data that are shown in the new supplementary Fig. 8: We prepared additional animals and analyzed them 1 day after DHT treatment. At this early time point, DHT dramatically reduced the number of 5HT-positive cells in all brainstem raphe nuclei and their axons in the spinal cord (Fig. 8), while CST axons remained unaffected (Fig. 5 b). Here BDA tracing of CST axons was performed 1 week after DHT treatment. Although 5-HT-positive neurons were already almost entirely lost 1 day after DHT treatment, BDA injection into the motor cortex resulted in a normal uptake and axonal transport in corticospinal axons. This resulted in an average of 2054 labeled axons/mm², which is comparable to animals without DHT-treatment (Fig. S2 b). Thus, cortical motor neurons and their axons were not affected by DHT and still intact.

Reviewer #3 How is the Raphe nucleus being defined anatomically? As the authors have mentioned, the raphe nucleus extends rostrocaudally; is a specific region chosen?

Response: The raphe nuclei were anatomically defined according to the Allen Brain Atlas and the pattern of 5-HT positive neurons. As landmarks, we used the pyramidal decussation, and the pyramidal tract in general, which are well visible as dark areas in the sections without staining (see crosssection below).

Thus, we did not selectively analyze specific regions but rather analyzed the whole Raphe nuclei instead. To address this point, we show, e.g., Fig. S12 A, B, a transverse scan through cleared tissue of the whole rostrocaudal extension of the raphe nuclei (indicated by 5-HT staining in blue). Co-staining demonstrates that pSTAT3-positive raphe neurons are distributed over the whole area of the nuclei. Moreover, we included the new data (Fig. 6 d-u) exemplary, showing the nucleus raphe magnus and the nucleus raphe pallidus. For Western blot analysis in Fig. 6 b, c, we also isolated the entire rostrocaudal length of the raphe nuclei above the pyramidal tracts.

Reviewer #3 Figure 6 - Transneuronal transduction of hIL-6 is the highlight of the paper. Therefore this warrants a more thorough quantitation. For example, the number of transduced cells (yellow cells) should be quantified in the RN? It appears that NRM doesn't have any transduced cells; is there a regional preference since the raphe spans a modest distance rostrocaudally?

Response: This comment is likely based on a misunderstanding as not transneuronal transduction, but transneuronal delivery of the hIL-6 protein is the focus of the paper. Transneuronal transduction can not occur via AAV2 delivery. Instead, virally expressed hIL-6 is transported along the axons of AAV2-transduced cortical neurons and released as a protein to stimulate raphe nuclei. This enables a relatively high level of transneuronal STAT3 activation in ~50% of all raphe neurons (Fig. 6, supplementary Fig. 12 b). This number is the appropriate readout for the efficiency of our treatment and given in the text.

In contrast to AAV2, AAV1 can transneuronally transduce neurons anterogradely and retrogradely (Aschauer et al., 2013; B et al., 2017; Jeong et al., 2016; Tervo et al., 2016), however, with moderate efficiency because AAVs are not able to reproduce themselves in the neuron. Thus, virus particles have to maintain intact after entering the cell and to be transported along the axons for a synaptic release. As a consequence, only some, but not all neurons are transneuronally transduced. We, therefore, used AAV1 only to express Cre in ROSA-reporter- mice to induce RFP expression and to verify, as a proof of principle, the connection between cortical

neurons and the raphe nuclei (Fig. 7 in the revised manuscript). In case the information might have been missed, we would like to point out that Fig. 7 b, c, e, and Fig. 7 b, d, f. show the expression of RFP in the NRM and NRPa. These photos show single sections, but we also detected many other positive cells in consecutive sections over the entire rostrocaudal area, which is now stated in the text. Thus, we did not detect any regional preference.

To make this point clearer, we added the following statements in the discussion on page 16: "... Moreover, we observed transneuronal transduction over the whole rostrocaudal length of the raphe nuclei, even though the overall transneuronal transduction rate was expectedly low. This is due to the specific properties of AAVs, which are not able to reproduce themselves without specific helper plasmids. Hence, virus particles have to remain intact after entering the cell and have to be directly transported along the axons for later synaptic release, which is a rather rare event. ... In contrast to AAV1 mediated transneuronal transduction, hIL-6 is continuously expressed, transported, and eventually released at axon terminals to secondary neurons as reflected by a high percentage of pSTAT3 positive serotonergic neurons (~50%) in all brain stem raphe nuclei."

Reviewer #3: Additionally, since the projections of cortical motoneurons into the raphe are being described here for the first time, other approaches to cross-verify these projections should be utilized. Retroviruses is a simple example of one approach.

Response: We followed this suggestion and successfully performed additional retrograde transsynaptic transduction of cortical motoneurons after viral injection into the raphe nuclei (Fig. 7 g-l).

Reviewer #3: In 6b-d, the green background is very high, the colocalization is barely distinguishable. It is really not clear what one is looking at. Green axons in the RN are visible, but the yellow, blue and red arrows are confusing and have not been explained. Panel B seems unnecessary.

Response: We agree that the background is relatively high. For this reason, the arrows were included. Arrows of one color follow the path of one single GFP positive axon. Sprouts are marked with arrows of the same color. Knowing this allows identification of the GFP labeled axons projecting into the raphe nuclei with 5-HT-positive neurons and axons (red). It was challenging to reduce the background in these scans through cleared wholemount tissue without losing the actual signal.

For this reason, we transferred this figure to the supplementary material (Fig. S 11) and included GFP staining in medulla-crosssections in Fig. 6 d-u. Here, the GFP-positive axons that project to the raphe nuclei are better visible. However, due to tissue sectioning, the axons can only be seen as fragments, but not in the entire length. Nevertheless, having both panels strongly supports our point.

Reviewer #3: Supp Fig 1- fewer arrows in b-e. Panel F should be quantified and IL6 staining should be shown in GFP control group for comparison. This is absolutely necessary to determine if IL6 is in fact increased in the hIL6 group.

Response: We followed this suggestion and reduced the number of arrows. We included IL-6 staining for the GFP group, which did not show any IL-6 positive cells. (supplementary Fig. 1 g). For quantification, we decided not to simply count the positive cells but operated an additional cohort of hIL-6 (n=7) and GFP (n=8) animals and quantified hIL-6 expression via Western blots 3 weeks after viral application (Fig. S1 h, i). Using this method, we were additionally able to prove that we expressed hIL-6, which runs at ~70 kDa (as a fusion protein of IL-6 and the receptor) instead of IL-6, which would be ~20 kDa.

Reviewer #3: The authors mention that no BDA-labelled axons were seen past 11mm and that confirms the completeness of the injury. This sounds like a good measure, but should be included in the supplemental figure.6.

Response: We followed this suggestion and included the additional data as new supplementary Fig. 5.

Reviewer #3: Figure 7G shows RN as the red nuclei, but previous figures refer to Raphe Nucleus as RN. Please be consistent.

Response: We changed the reference to RedN instead of RN for the red nucleus. These panels have been moved to supplementary Fig. 12 c, d.

Reviewer #3: Figure 2i has huge error bars which is concerning especially for the PTEN^{-/-} group. Can this be explained?

Response: This variability is likely caused by the relatively small values of regenerating axons since CST regeneration is the most refractive among all spinal tracts. As shown, only relatively small numbers of regenerating axons were quantified, in some cases, almost no regeneration. Thus, a higher error-bar relative to the average value is not unusual and a general issue for small values. In this context, it is worth mentioning that PTEN^{-/-} induced regeneration highly depends on the width of the lesion site, as reported by Zukor et al. 2013, which of course, slightly varies from animal to animal in single groups. However, whether this has contributed to the variability beyond the “small value effect,” we don’t know.

Nevertheless, the longest axons seen in PTEN^{-/-} animals were around 2 mm, while we saw no regenerating axons in controls. This finding was very consistent with Liu et al. and Geoffrey et al., and in hIL-6-treated animals, the average length was significantly longer (up to 6 mm). Moreover, our paper unambiguously demonstrates that CST regeneration does not mainly contribute to the functional effects observed in hIL-6 treated animals. If the editors feel that these general explanations need to be included in the manuscript, we could do that, but believe that this is not necessary because of the general phenomenon of measuring small effects.

Reviewer #3: Figure 4- the authors inject hIL-6 bilaterally but don’t see any improvement in regeneration, which is counter-intuitive. The authors should mention relevant information about the percentage of ipsi and contralateral projections for these tracts, which will be useful for readers in thinking about these results and explain the rationale behind why a bilateral injection was tested.

Response: The rationale of this experiment was our finding that CST regeneration did not mainly contribute to functional recovery and that we observed similar raphespinal axon regeneration (Fig. 4c), and then similar functional recovery (Fig. 3 f) as compared to bilateral cortical injection. We showed that axon collaterals of one motor cortex hemisphere project into both hemispheres of the raphe nuclei along the whole rostrocaudal length (supplementary Fig. 11). The projection of axons to both sides of the raphe nuclei is most likely because they are located in the medial column and, therefore, in the center of the brainstem. Very accurate quantification of axon tips based on these photos is difficult. More relevant are, however, the secondary neurons with activated STAT3 (pSTAT3), which is direct evidence that neurons in both sides of the RN were affected. As stated and shown in the paper, unilateral AAV2 injection into the left motor cortex caused STAT3 activation in equal numbers of neurons in both hemispheres of the raphe nuclei (new Fig. 6 d-u)). We now quantified the percentage of pSTAT3-positive neurons on each side. Exact percentages are now provided in the text of the result section.

Reviewer #3: Figure 5F should include a legend to indicate which group is in red and black. The statistical significance is shown within each group; however, the statistical significance between gfp and hIL-6 groups is also important and should be shown or mentioned in the figure legend at the very least, especially for 6w1d time point?

Response: We followed these suggestions and included a legend for Fig. 5 f in the revised version of the manuscript. We also indicate the statistics between gfp and hIL-6 in this figure. The differences are highly significant 6 weeks after SCC ($p=0.00053$), whereas 1 day after DHT, the difference between gfp and hIL-6 is not significant ($p=0.12$) anymore, supporting a strong effect of DHT on hIL-6 mediated functional recovery.

Reviewer #3: Supp Fig7H is unclear. Is this a horizontal section??

Response: Supplementary Fig. 7 h is now supplementary Fig. 12 b in the revised version of the paper and represents a transverse/horizontal scan through a Z-plane of 100 μm from cleared wholemount tissue of the brainstem scanned from the ventral side above the pyramidal tract as indicated by the dotted box in panel A. The scan spans the whole rostrocaudal length of the brain stem raphe nuclei. This information is now stated in the figure legend.

Reviewer #3: The methods section has a paragraph describing Intraspinal injections of hIL-6. Perhaps I missed it, but I didn't see intraspinal injections being described in any of the figures or main text. Please check thoroughly.

Response: Thank you for this comment. This section was indeed included accidentally. We removed it in the revised manuscript.

Literature:

- Aschauer, D.F., Kreuz, S., and Rumpel, S. (2013). Analysis of transduction efficiency, tropism and axonal transport of AAV serotypes 1, 2, 5, 6, 8 and 9 in the mouse brain. *PLoS One* 8, e76310.
- B, Z., XL, C., ZG, Z., L, M., F, L., HW, T., and LI, Z. (2017). AAV-Mediated Anterograde Transsynaptic Tagging: Mapping Corticocollicular Input-Defined Neural Pathways for Defense Behaviors. *Neuron* 93.
- Geoffroy, C.G., Lorenzana, A.O., Kwan, J.P., Lin, K., Ghassemi, O., Ma, A., Xu, N., Creger, D., Liu, K., He, Z., *et al.* (2015). Effects of PTEN and Nogo Codeletion on Corticospinal Axon Sprouting and Regeneration in Mice. *J Neurosci* 35, 6413-6428.
- Heskamp, A., Leibinger, M., Andreadaki, A., Gobrecht, P., Diekmann, H., and Fischer, D. (2013). CXCL12/SDF-1 facilitates optic nerve regeneration. *Neurobiology of disease* 55, 76-86.
- Jeong, M., Kim, Y., Kim, J., Ferrante, D.D., Mitra, P.P., Osten, P., and Kim, D. (2016). Comparative three-dimensional connectome map of motor cortical projections in the mouse brain. *Sci Rep* 6, 20072.
- Leibinger, M., Andreadaki, A., Diekmann, H., and Fischer, D. (2013). Neuronal STAT3 activation is essential for CNTF- and inflammatory stimulation-induced CNS axon regeneration. *Cell Death Dis* 4, e805.
- Leibinger, M., Andreadaki, A., and Fischer, D. (2012). Role of mTOR in neuroprotection and axon regeneration after inflammatory stimulation. *Neurobiology of disease* 46, 314-324.
- Leibinger, M., Andreadaki, A., Gobrecht, P., Levin, E., Diekmann, H., and Fischer, D. (2016). Boosting Central Nervous System Axon Regeneration by Circumventing Limitations of Natural Cytokine Signaling. *Mol Ther* 24, 1712-1725.
- Leibinger, M., Andreadaki, A., Golla, R., Levin, E., Hilla, A.M., Diekmann, H., and Fischer, D. (2017). Boosting CNS axon regeneration by harnessing antagonistic effects of GSK3 activity. *Proc Natl Acad Sci U S A* 114, E5454-E5463.
- Leibinger, M., Hilla, A.M., Andreadaki, A., and Fischer, D. (2019). GSK3-CRMP2 signaling mediates axonal regeneration induced by Pten knockout. *Commun Biol* 2, 318.
- Levin, E., Diekmann, H., and Fischer, D. (2016). Highly efficient transduction of primary adult CNS and PNS neurons. *Sci Rep* 6, 38928.
- Levin, E., Leibinger, M., Gobrecht, P., Hilla, A., Andreadaki, A., and Fischer, D. (2019). Muscle LIM Protein Is Expressed in the Injured Adult CNS and Promotes Axon Regeneration. *Cell reports* 26, 1021-1032 e1026.
- Liu, K., Lu, Y., Lee, J.K., Samara, R., Willenberg, R., Sears-Kraxberger, I., Tedeschi, A., Park, K.K., Jin, D., Cai, B., *et al.* (2010). PTEN deletion enhances the regenerative ability of adult corticospinal neurons. *Nat Neurosci* 13, 1075-1081.
- Rognum, I.J., Haynes, R.L., Vege, Á., Yang, M., Rognum, T.O., and Kinney, H.C. (2009). Interleukin-6 and the serotonergic system of the medulla oblongata in the sudden infant death syndrome. *Acta Neuropathol* 118, 519-530.
- Tervo, D.G., Hwang, B.Y., Viswanathan, S., Gaj, T., Lavzin, M., Ritola, K.D., Lindo, S., Michael, S., Kuleshova, E., Ojala, D., *et al.* (2016). A Designer AAV Variant Permits Efficient Retrograde Access to Projection Neurons. *Neuron* 92, 372-382.
- Watanabe, D., Yoshimura, R., Khalil, M., Yoshida, K., Kishimoto, T., Taga, T., and Kiyama, H. (1996). Characteristic localization of gp130 (the signal-transducing receptor component used in common for IL-6/IL-11/CNTF/LIF/OSM) in the rat brain. *The European journal of neuroscience* 8, 1630-1640.
- Zukor, K., Belin, S., Wang, C., Keelan, N., Wang, X., and He, Z. (2013). Short hairpin RNA against PTEN enhances regenerative growth of corticospinal tract axons after spinal cord injury. *J Neurosci* 33, 15350-15361.

Reviewers' Comments:

Reviewer #2:

Remarks to the Author:

In the revised version of the manuscript Leibinger et al included many more experimental data which overall support their initial claims and address most of my concerns sufficiently.

So overall I am happy to support publication of the manuscript as it stands.

Reviewer #3:

Remarks to the Author:

Thank you for carefully addressing all the comments, especially during these difficult times.

I think it is worthwhile mentioning in the text the details regarding animals that were excluded. I don't think the data needs to be shown in supplemental figures. The reason being, with these types of studies, it is always useful to understand the exclusion criteria and how to interpret the findings.

All other concerns have been satisfactorily addressed.

Response to the reviewers' comments:

REVIEWERS' COMMENTS

Reviewer #2 (Remarks to the Author): In the revised version of the manuscript Leibinger et al included many more experimental data which overall support their initial claims and address most of my concerns sufficiently. So overall I am happy to support publication of the manuscript as it stands.

RESPONSE: Thank you

Reviewer #3 (Remarks to the Author): Thank you for carefully addressing all the comments, especially during these difficult times. I think it is worthwhile mentioning in the text the details regarding animals that were excluded. I don't think the data needs to be shown in supplemental figures. The reason being, with these types of studies, it is always useful to understand the exclusion criteria and how to interpret the findings. All other concerns have been satisfactorily addressed.

RESPONSE: Thank you. We followed this suggestion and mention the animals in the M&M part. Changes are indicated in red letter in the manuscript.